# AgilePruner: An Empirical Study of Attention and Diversity for Adaptive Visual Token Pruning in Large Vision-Language Models

**Changwoo Baek**[1][*]   **Jouwon Song**[2][*]   **Sohyeon Kim**[1][*]   **Kyeongbo Kong**[1][†]
[1]Pusan National University, {higok18, shkim0503, kbkong}@pusan.ac.kr
[2]LG Electronics, juwon05.song@lge.com

## Abstract

Large Vision-Language Models (LVLMs) have adopted visual token pruning strategies to mitigate substantial computational overhead incurred by extensive visual token sequences. While prior works primarily focus on either attention-based or diversity-based pruning methods, in-depth analysis of these approaches' characteristics and limitations remains largely unexplored. In this work, we conduct thorough empirical analysis using effective rank (erank) as a measure of feature diversity and attention score entropy to investigate visual token processing mechanisms and analyze the strengths and weaknesses of each approach. Our analysis reveals two insights: (1) Our erank-based quantitative analysis shows that many diversity-oriented pruning methods preserve substantially less feature diversity than intended; moreover, analysis using the CHAIR dataset reveals that the diversity they do retain is closely tied to increased hallucination frequency compared to attention-based pruning. (2) We further observe that attention-based approaches are more effective on simple images where visual evidence is concentrated, while diversity-based methods better handle complex images with distributed features. Building on these empirical insights, we show that incorporating image-aware adjustments into existing hybrid pruning strategies consistently improves their performance. We also provide a minimal instantiation of our empirical findings through a simple adaptive pruning mechanism, which achieves strong and reliable performance across standard benchmarks as well as hallucination-specific evaluations. Our project page available at https://cvsp-lab.github.io/AgilePruner.

## 1 Introduction

Large Vision-Language Models (LVLMs) (Liu et al., 2023; Wang et al., 2024; Liu et al., 2024a) have garnered significant attention by integrating various modalities such as images, text, and video to achieve human-level vision-language reasoning capabilities. In particular, visual information is encoded into token embeddings that can be processed by language models, generating hundreds of visual tokens in this process. The increase in the number of these tokens causes the complexity of attention-based computations to scale quadratically, significantly impacting inference speed and efficiency.

To address these issues, numerous researchers have attempted to reduce computational costs by removing unnecessary or redundant visual tokens through token pruning methods (Xing et al., 2024; Chen et al., 2024; Zhang et al., 2025b). These existing methods typically employ two main methods. The first is attention-based methods (Zhang et al., 2024; Yang et al., 2025; Arif et al., 2025), which consider tokens with high attention scores as important information and remove the rest. The second is diversity-based methods (Alvar et al., 2025), which reduce redundancy based on feature similarity between visual tokens. Each approach exhibits distinct tendencies. Attention-based methods prioritize the preservation of highly weighted tokens, which can result in concentrated but sometimes repetitive selections. In contrast, diversity-based methods encourage broader coverage,

---

[*]Equal contribution
[†]Corresponding author

often at the cost of overlooking important tokens. There have also been attempts to combine these two strategies through hybrid schemes (Zhang et al., 2025a; Shang et al., 2025).

Despite the emergence of attention-based, diversity-based, and hybrid token-pruning strategies, their actual behaviors remain insufficiently characterized. In particular, (i) how much feature-space diversity these methods truly preserve and (ii) how the retained-token properties influence hallucination tendencies in LVLMs have not been systematically examined. Furthermore, (iii) it remains unclear whether different image types naturally favor attention-based or diversity-based pruning strategies.

To clarify these aspects, we conduct a two-part empirical analysis. First, we characterize the intrinsic behaviors of existing pruning paradigms—quantifying their retained diversity using effective rank (erank) (Roy & Vetterli, 2007) and examining how this relates to hallucination patterns across image types. Second, we analyze how pruning effectiveness shifts with image-level complexity, revealing when each paradigm is preferable.

Using erank and attention-entropy, our analysis shows that:

- **(Method-level behavior: diversity & hallucination)** Many diversity-aware pruning methods preserve substantially less diversity than intended. More importantly, higher retained diversity is strongly associated with increased hallucination frequency (CHAIR (Rohrbach et al., 2018)), whereas attention-based pruning—which retains lower-diversity token sets—produces more conservative outputs with suppressed hallucinations.
- **(Image-level behavior: complexity-dependent preference)** Attention-based pruning is more effective on simple images where essential cues are concentrated in a small number of tokens, while diversity-based pruning excels on complex images where semantic information is widely distributed.

Leveraging these empirical findings, we next examine whether the observed behaviors can be translated into practical improvements. By applying the identified tendencies to existing pruning strategies—including hybrid approaches as well as direct mixtures of attention- and diversity-based methods—we incorporate a simple image-aware adjustment derived from our analysis. This adjustment consistently improves performance across benchmarks, suggesting that the empirical patterns uncovered in our study are robust and broadly applicable.

We further introduce a simple threshold-based pruning procedure that operationalizes the empirical behaviors identified above. The method explores tokens in descending attention order and removes redundant tokens based on similarity, using an adaptively set threshold informed by image-level complexity. Although intentionally minimal in design, this instantiation achieves strong performance across nine standard datasets—often matching or surpassing existing pruning methods—and, consistent with our empirical analysis, it further mitigates hallucination tendencies as validated on the CHAIR benchmark. These results show that the empirical principles revealed in our study are not only explanatory but also practically effective. Moreover, we observe the same behavioral trends and improvements across larger and architecturally different LVLMs, including LLaVA-1.5-13B, LLaVA-NeXT-7B, and Qwen2.5-VL-7B, indicating that the discovered principles are robust and model-agnostic.

In summary, our contributions are three-fold:

- We provide the first erank-based characterization of how existing pruning methods preserve feature diversity and how this retained diversity relates to hallucination behavior.
- We reveal a consistent image-complexity–dependent preference between attention-based and diversity-based pruning, explaining when each paradigm succeeds or fails.
- We show that these empirical principles are actionable by improving existing pruning methods and by presenting a minimal adaptive instantiation that achieves strong, consistent performance.

## 2  RELATED WORKS

**Large Vision-Language Models**   The advances in Large Language Models (LLMs) (Bai et al., 2023; Touvron et al., 2023; Yu et al., 2024; Cai et al., 2024) have extended to LVLMs (Liu et al.,

2023; Wang et al., 2024; Liu et al., 2024a), which are now widely applied across domains for reason over diverse modalities. Among them, LVLMs specialized for vision–language integration, have drawn particular attention. A typical LVLM architecture comprises a vision encoder (Radford et al., 2021; Tschannen et al., 2025), a modality projector, and an LLM. The vision encoder converts input images into visual tokens, and the projector aligns these tokens with the LLM's word-embedding space so that the LLM can effectively interpret and process visual information. However, visual tokens are not only numerous but also highly redundant—a tendency that becomes even more pronounced with high-resolution images or video inputs. Combined with the autoregressive nature of the LLM decoder, this leads to a substantial drop in inference efficiency.

**Visual token reduction**    Reducing redundant and unnecessary visual tokens is an effective way to decrease computation and memory usage, thereby improving the inference efficiency of LLMs. In particular, many studies have adopted token pruning methods that require no additional training, and these methods can largely be categorized as follows. **(i) Attention-based method:** These methods prune visual tokens by leveraging the attention distribution in the penultimate layer of the vision encoder before the tokens are fed into the LLM (Zhang et al., 2024; Shang et al., 2025; Yang et al., 2025; Zhang et al., 2025a; Arif et al., 2025). Based on the observation that image information in the vision encoder tends to concentrate on a small set of key tokens, these methods utilize attention scores from the output layer to select a limited number of tokens that aggregate global information. However, they tend to retain similar tokens concentrated in specific regions, which results in insufficient diversity to fully represent the entire token set. **(ii) Diversity-based method:** These methods leverage inter-token similarity to enhance the diversity of the selected token set (Alvar et al., 2025), thereby encouraging the selection of more diverse tokens. However, they introduce additional computational overhead and risk discarding important tokens. There have been recent efforts to merge these two strategies through hybrid pruning schemes (Zhang et al., 2025a; Shang et al., 2025). These approaches represent the dominant strategies for pre-pruning, and alongside them, there has also been increasing interest in dynamically determining the number of tokens to retain, as explored in recent work such as ATP-LLaVA (Ye et al., 2025).

## 3    PRELIMINARIES

**Visual token pruning.**    Since vision encoders often produce hundreds of tokens, passing all tokens into the LLM incurs heavy computational cost. Visual token pruning addresses this issue by removing redundant tokens and retaining only $K \ll N$ tokens. Formally, pruning defines a function $g$ such that $\tilde{V}' = g(\tilde{V}, K)$, where $\tilde{V}'$ denotes the retained subset. Among existing methods, we focus on two categories: *(i) attention-based*, which select tokens according to [CLS] token attention scores, and *(ii) diversity-based*, which aim to reduce redundancy and preserve feature diversity. These strategies provide the foundation for our subsequent empirical analysis. Fig. 1 provides an overview of our analytical framework, which evaluates the vision encoder from two perspectives: attention concentration and token embedding diversity.

**Attention concentration via attention entropy.**    To assess the concentration of attention within the vision encoder, we compute the Shannon entropy of the class token's attention score. Given the head-averaged attention score $\alpha \in \mathbb{R}^N$ obtained from the penultimate layer, we exclude the self-attention score of the class token and renormalize the remaining score into a valid probability distribution:

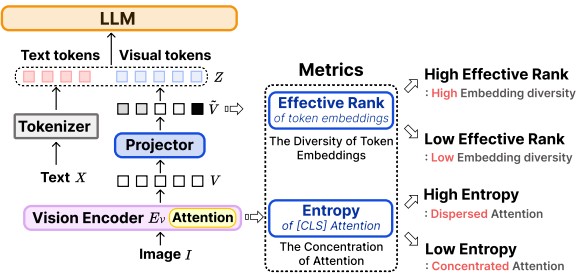

Figure 1: Overview of attention entropy and erank.

$$p_i = \frac{\alpha_i}{\sum_{j \neq \text{CLS}} \alpha_j}, \quad \sum_i p_i = 1. \quad (1)$$

The entropy is then computed as

$$H(p) = -\sum_i p_i \log p_i. \quad (2)$$

The entropy value $H(p)$ quantifies how attention is distributed across tokens: a lower value indicates that the class token attends strongly to a few regions, whereas a higher value suggests a more uniform

distribution over multiple visual tokens. We refer to this measure as **attention entropy** in the rest of this paper.

**Token embedding diversity via erank.** To quantitatively assess the diversity of token embeddings, we adopt the notion of erank (Roy & Vetterli, 2007). Unlike the conventional matrix rank, the erank is an entropy-based measure that evaluates the number of dimensions effectively utilized by a matrix.

Given a token embedding matrix $A \in \mathbb{R}^{N \times d_l}$, we first obtain its singular values $\{\sigma_i\}$ via singular value decomposition (SVD). Let

$$L = \min(N, d_l), \quad q_i = \frac{\sigma_i}{\sum_{j=1}^{L} \sigma_j}, \quad q_i \in \mathbb{R}^L. \tag{3}$$

The erank is then defined as

$$\mathrm{erank}(A) = \exp\left( -\sum_{i=1}^{L} q_i \log q_i \right).$$

The value of $\mathrm{erank}(A)$ ranges between 1 and $L$. A low erank indicates that the embedding representation is concentrated in a few dominant dimensions, whereas a high erank suggests that the embedding space is more evenly distributed across multiple dimensions.

## 4 EMPIRICAL STUDIES

This section presents empirical analyses of attention-based and diversity-based token pruning methods. We focus on two aspects: the diversity preserved by existing methods and its relation to hallucinations (Sec. 4.1) the impact of image complexity on token selection strategies (Sec. 4.2) . Building on the insights from these two analyses, we then introduce simple adaptive pruning framework (Sec. 4.3).

### 4.1 EMPIRICAL ANALYSIS OF ATTENTION-BASED AND DIVERSITY-BASED PRUNING

To better understand the intrinsic behaviors of existing pruning paradigms, we conduct an empirical study using erank, which captures the diversity of token-level visual features. By analyzing these signals on the token sets actually selected by different pruning methods, we aim to characterize how much semantic diversity each approach preserves and how these differences in diversity ultimately influence the behavior as revealed through hallucination analysis.

#### 4.1.1 ANALYZING DIVERSITY PRESERVATION IN EXISTING PRUNING PARADIGMS VIA ERANK

**Quantitative Comparison of Diversity Preservation via erank** We begin by assessing how well existing pruning methods preserve semantic diversity by measuring erank (see Table. 1). This comparison reveals a clear

|  | PruMerge+ | VisionZip | VisPruner | DivPrune |
|---|---|---|---|---|
| **Erank** | 10.91 | 14.02 | 14.35 | **21.84** |

Table 1: Mean erank of retained 64 tokens on POPE.

structure across approaches. DivPrune (Alvar et al., 2025) achieves the highest mean erank (21.84), reflecting its explicit objective of maximizing geometric dispersion among tokens. At the other end of the spectrum, PruMerge+ (Shang et al., 2025) records the lowest erank (10.91), indicating limited diversity under its spatial-sampling strategy. VisPruner (Zhang et al., 2025a) (14.35) and VisionZip (Yang et al., 2025) (14.02) form a mid-range group with similar diversity levels—higher than PruMerge+ but still lower than that of DivPrune. These results show that, even under the same token budget, different pruning strategies retain markedly different degrees of semantic diversity.

**Relative Strengths of Diversity Mechanisms Across Methods** A closer examination of these methods clarifies this pattern. Although VisPruner, VisionZip, and PruMerge+ all adopt a strategy that prioritizes tokens with high attention scores, the strength of their auxiliary diversity mechanisms

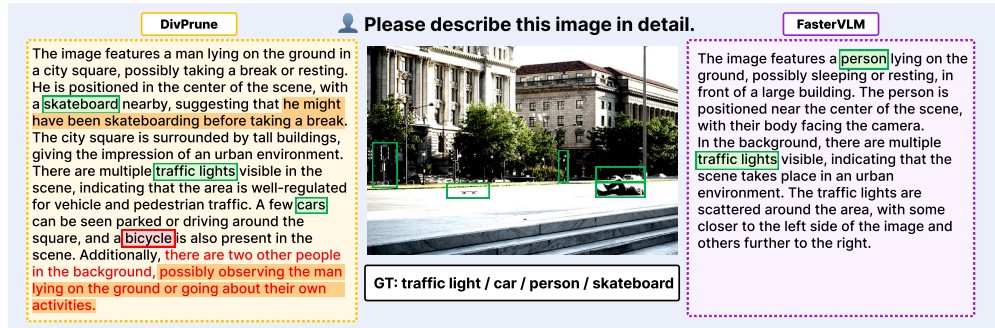

**Figure 2: Response patterns of DivPrune (diversity-based) vs. FasterVLM (attention-based).** DivPrune's responses are more comprehensive but risk hallucination, whereas FasterVLM produces safer, more focused descriptions. In the annotations, ■GT Obj. and ■Hallucinated Obj. label object words; ■marks DivPrune-specific phrasing; **red text** indicates incorrect phrases.

differs substantially. VisPruner removes feature-redundant tokens before applying attention filtering, providing a more direct form of diversity preservation and resulting in slightly higher erank among the attention-driven approaches. In contrast, VisionZip's token merging and PruMerge+'s spatial sampling introduce only limited dispersion, leading to lower overall diversity. Nevertheless, all three remain fundamentally constrained by their reliance on attention-guided token selection, which inherently limits the semantic diversity they can retain compared to a method like DivPrune, whose primary objective is to maximize geometric dispersion.

### 4.1.2 THE RELATIONSHIP BETWEEN PRUNING METHODS AND HALLUCINATION

Object hallucination occurs frequently in LVLMs and is a critical issue that undermines their reliability. In this section, we compare and analyze the characteristics of attention score–based and diversity-based methods from the perspective of hallucination, aiming to identify how the two pruning methods differ in inducing hallucinations. To this end, we evaluate object hallucination in the LLaVA-1.5-7B model using not only the datasets commonly employed in prior token reduction studies but also additional datasets, and we present the corresponding results.

**Object hallucination.** To assess the degree of object hallucination in the image captioning task, we employ the CHAIR dataset. CHAIR quantifies the proportion of objects mentioned in generated captions that are absent in the ground-truth annotations, providing two sub-metrics, $C_I$ and $C_S$, as defined in Eq. 5.

$$C_I = \frac{|\{\text{hallucinated objects}\}|}{|\{\text{all mentioned objects}\}|}, \qquad C_S = \frac{|\{\text{captions with hallucinated objects}\}|}{|\{\text{all captions}\}|}. \qquad (5)$$

Each metric evaluates hallucination at the instance level and the sentence level, respectively, and lower values indicate better performance. As auxiliary metrics, we also report recall and len, where recall denotes the proportion of ground-truth objects mentioned in the generated captions, and len represents the average number of words in the generated captions.

**Results on the CHAIR dataset.** Table 2 presents the results of the attention-based and diversity-based methods on the CHAIR dataset. Despite small differences in Len, the diversity-based methods exhibit higher values of the hallucination metrics $C_S$ and $C_I$ compared to the attention-based methods, suggesting that selecting diverse tokens with low feature similarity may increase the likelihood of hallucination. In contrast, the diversity-based methods achieve higher recall, indicating that they are able to capture a larger number of objects than the attention-based ones.

| Method | $C_s \downarrow$ | $C_i \downarrow$ | Recall↑ | Len |
|---|---|---|---|---|
| **LLaVA-1.5-7B** | 51.0 | 13.9 | 78.7 | 101.4 |
| *Attention-based methods* | | | | |
| FasterVLM (arXiv'24) | 45.4 | 13.5 | 69.3 | 94.0 |
| PruMerge+ (ICCV'25) | 45.2 | 15.6 | 66.7 | 91.4 |
| Vispruner (ICCV'25) | 49.8 | 15.0 | 72.6 | 96.7 |
| *Diversity-based methods* | | | | |
| DivPrune (CVPR'25) | 57.4 | 18.0 | 76.4 | 101.1 |
| FPSPruner[†] | 58.6 | 18.6 | 76.0 | 100.5 |

Table 2: **Comparison on CHAIR.** [†]FPSPruner is based on farthest point sampling (FPS), which iteratively selects the farthest token to guarantee diversity.

**Comparison of the two methods in terms of response patterns.** The two methods also differ in their response patterns. Fig. 2 illustrates this distinction, showing that the diversity-based method

Figure 3: **Diversity vs. attention in pruning across datasets and image complexities.** (a) High-erank methods perform better on complex datasets (POPE), while low-erank methods excel on simple datasets (ScienceQA). (b) Simple images show low entropy and erank, leading to concentrated attention suitable for attention-based pruning. Complex images show high entropy and erank, where diversity-based pruning becomes more effective.

DivPrune generates broader and more open-ended descriptions and often includes speculative expressions, as highlighted in yellow. In addition, the diversity-based response refers to a larger set of ground-truth objects marked in green, but at the same time it also introduces hallucinated objects highlighted in red and incorrect phrases emphasized in red text. In contrast, the attention-based method FasterVLM focuses on the main objects and provides more conservative and reliable explanations, thereby suppressing hallucinations that frequently appear in diversity-based outputs.

**Effect of attention-based selection on object hallucination.** Based on the observation that attention-based token selection tends to reduce hallucination, we conducted experiments to quantitatively assess the effect of attention-based selection on hallucination by varying the balance between diversity-based and attention-based selection.

| Method | $C_s \downarrow$ | $C_i \downarrow$ | Recall↑ | Len | Mean erank | Mean attn. |
|--------|------|------|--------|------|-----------|-----------|
| | | | *Retain 64 Tokens* | | | |
| R=0 | 57.4 | 18.0 | 76.4 | 101.1 | 21.14 | 0.0035 |
| R=0.25 | 50.8 | 16.8 | 74.5 | 97.6 | 14.98 | 0.0065 |
| R=0.50 | 46.2 | 14.5 | 73.7 | 95.5 | 14.38 | 0.0072 |
| R=0.75 | 45.2 | 14.1 | 70.5 | 94.0 | 13.58 | 0.0076 |

Table 3: **Effect of attention-based selection ratio $R$ on CHAIR metrics.** Higher attention-based selection reduces hallucination ($C_S$, $C_I$) but lowers recall.

As shown in Table 3, we fixed the token budget at 64 and gradually reduced the number of tokens selected by the DivPrune, while replacing them with tokens having higher attention scores. The experimental results demonstrate that increasing the proportion of attention-based selection leads to a gradual decrease in the hallucination metrics $C_S$ and $C_I$.

*These findings suggest that selecting tokens solely based on diversity can induce relatively higher hallucination, whereas tokens with high attention scores, which concentrate critical information, play a pivotal role in generating reliable captions and mitigating hallucination.*

## 4.2 ANALYZING SAMPLE-DEPENDENT EFFECTS OF PRUNING STRATEGIES

We previously examined how different pruning methods vary in the degree of semantic diversity they preserve, as measured by erank, and how these differences relate to phenomena such as hallucination. Building on these observations, we now investigate how such characteristics influence model behavior in downstream reasoning tasks.

**When diversity helps and when attention-based selection prevails.** Our experiments show that neither diversity-based pruning nor attention-based pruning is universally superior. As illustrated in Fig. 3 (a), we observe a clear dataset-dependent pattern: methods that retain more diverse tokens (i.e., higher erank) tend to perform better on datasets such as POPE, whereas attention-based approaches achieve higher accuracy on datasets like ScienceQA. Motivated by these contrasting trends, we further analyze which types of image samples favor diversity-based methods and which benefit more from attnention-based token selection.

**Image complexity affects attention entropy and diversity.** We first analyzed how image complexity affects LVLMs. To this end, we measured the concentration of attention using attention entropy and assessed the diversity of token features using erank. The analysis was conducted on LLaVA-v1.5-7B using the MME Benchmark (Yin et al., 2024), where tasks such as *OCR, Numerical calculation, and Text translation* involve images with plain backgrounds and few key objects, whereas tasks such as *Position, Scene, and Count* involve images with mixed backgrounds and mul-

| Method | MME | | | ScienceQA |
|---|---|---|---|---|
| | OCR | Numerical Cal. | Text Translation | |
| Metric | | | | |
| Att. entropy | 4.61 | 4.47 | 4.39 | 4.45 |
| Erank | 78 | 58 | 49 | 74 |
| Scores after pruning 576 → 64 tokens | | | | |
| **Att. based** | **140** | **55** | **100** | **69.51** |
| Div. based | 130 | 40 | 80 | 67.53 |

(a) Results on datasets with **simple images.**

| Method | MME | | | POPE |
|---|---|---|---|---|
| | Position | Scene | Count | |
| Metric | | | | |
| Att. entropy | 4.90 | 4.86 | 4.82 | 4.87 |
| Erank | 109 | 103 | 102 | 106 |
| Scores after pruning 576 → 64 tokens | | | | |
| Att. based | 105 | 157 | 120 | 77.4 |
| **Div. based** | **111** | **168** | **140** | **86.0** |

(b) Results on datasets with **complex images.**

Table 4: **Attention entropy and erank on simple and complex image datasets.** Simple images exhibit lower entropy and erank, while complex images show higher values, and the two pruning methods show contrasting performance between simple and complex images.

| Method | GQA | SQA$^{IMG}$ | POPE | MME | Rel. |
|---|---|---|---|---|---|
| *All 576 Tokens* | | | | | |
| LLaVA-1.5-7B | 61.9 | 69.5 | 85.9 | 1862 | 100.0% |
| *Retain 128 Tokens* | | | | | |
| BAT (CVPR'23) | 58.6 | 69.3 | 85.3 | 1737 | 96.75% |
| BAT + Inverse | 58.4$_{-0.2}$ | 69.1$_{-0.2}$ | 85.0$_{-0.3}$ | 1734$_{-3}$ | 96.46%$_{-0.19}$ |
| **BAT + Ours** | **58.8**$_{+0.2}$ | **69.4**$_{+0.1}$ | **86.8**$_{+0.9}$ | **1782**$_{+45}$ | **97.91%**$_{+1.11}$ |
| Vispruner (ICCV'25) | 58.2 | 69.0 | 84.6 | 1768 | 96.72% |
| Vispruner + Inverse | 57.9$_{-0.3}$ | 68.8$_{-0.2}$ | 85.3$_{+0.7}$ | 1744$_{-24}$ | 96.35%$_{-0.37}$ |
| **Vispruner + Ours** | **58.6**$_{+0.4}$ | **69.1**$_{+0.1}$ | **85.5**$_{+0.9}$ | **1787**$_{+19}$ | **97.32%**$_{+0.60}$ |
| *Retain 64 Tokens* | | | | | |
| BAT (CVPR'23) | 56.6 | 68.8 | 81.5 | 1683 | 93.91% |
| BAT + Inverse | 55.8$_{-0.8}$ | 68.7$_{-0.1}$ | 79.7$_{-1.8}$ | 1602$_{-81}$ | 91.98%$_{-1.93}$ |
| **BAT + Ours** | **56.9**$_{+0.3}$ | **69.1**$_{+0.3}$ | **83.5**$_{+2.0}$ | **1692**$_{+9}$ | **94.85%**$_{+0.94}$ |
| Vispruner (ICCV'25) | 55.4 | 69.1 | 80.4 | 1650 | 92.78% |
| Vispruner + Inverse | 55.2$_{-0.2}$ | 68.9$_{-0.2}$ | 78.0$_{-2.4}$ | 1643$_{-7}$ | 91.83%$_{-0.95}$ |
| **Vispruner + Ours** | **55.9**$_{+0.5}$ | **69.3**$_{+0.2}$ | **81.5**$_{+1.1}$ | **1671**$_{+21}$ | **93.76%**$_{+0.98}$ |

Table 5: Performance results of combining diversity- and attention-based pruning methods.

| Method | GQA | SQA$^{IMG}$ | POPE | MME | Rel. |
|---|---|---|---|---|---|
| *All 576 Tokens* | | | | | |
| LLaVA-1.5-7B | 61.9 | 69.5 | 85.9 | 1862 | 100.0% |
| *Retain 128 Tokens* | | | | | |
| Divprune (CVPR'25) | 59.4 | 68.6 | 87.0 | 1707 | 96.90% |
| FasterVLM (ArXiv'24) | 57.9 | 68.5 | 83.2 | 1757 | 95.80% |
| Divprune + FasterVLM | 58.4 | 68.5 | 84.9 | 1768 | 96.66% |
| **Divprune + FasterVLM (adaptive)** | 58.9 | 68.8 | 86.0 | 1787 | **97.55%** |
| *Retain 64 Tokens* | | | | | |
| Divprune (CVPR'25) | 57.5 | 68.0 | 85.5 | 1615 | 94.25% |
| FasterVLM (ArXiv'24) | 55.0 | 69.0 | 77.4 | 1665 | 91.91% |
| Divprune + FasterVLM | 56.8 | 68.6 | 82.2 | 1681 | 94.08% |
| **Divprune + FasterVLM (adaptive)** | 57.2 | 68.9 | 83.5 | 1690 | **94.87%** |

Table 6: Adaptive combination of attention- and diversity-based pruning outperforms fixed schemes.

tiple objects, making them relatively complex. In addition, we included *ScienceQA* (simple) and *POPE* (complex) from external benchmarks for a more comprehensive analysis. Indeed, as shown in Table 4, simple images exhibited lower attention entropy, as in the case of *OCR* (4.61), where the vision encoder could readily concentrate information into a few dominant regions. In contrast, complex images such as *POPE* reached higher values (4.87), indicating more dispersed attention across multiple regions. Consistently, simple images also showed lower erank, such as *OCR* (78), while complex images reached higher values, such as *POPE* (106), reflecting redundant versus diverse token representations.

**Performance divergence by image complexity.** Our analysis reveals that the performance of these two approaches diverges depending on dataset characteristics, as shown in Table 4. As a result, diversity-based methods outperformed in high erank tasks, while attention-based methods were superior in low erank tasks. This performance reversal appears to be driven by differences in image complexity. As illustrated in Fig. 3 (b), simple images allow the vision encoder to easily concentrate attention on specific regions, leading to concentrated information. In such cases, attention-based methods can effectively select these concentrated tokens. In contrast, complex images contain multiple objects and mixed backgrounds, causing information to be dispersed across the entire image. In this scenario, diversity-based methods that capture a broader range of features become more effective.

## 4.3 FROM EMPIRICAL INSIGHTS TO ADAPTIVE TOKEN PRUNING

**Analysis-driven enhancement of pruning methods.** Sections 4.1 and 4.2 revealed two consistent empirical patterns: (i) different pruning paradigms preserve distinct levels of feature diversity and exhibit characteristic hallucination behaviors, and (ii) image complexity systematically determines whether attention- or diversity-oriented selection is preferable. To examine whether these findings are practically useful, we first apply them to existing pruning strategies. Whereas hybrid methods such as VisPruner (Zhang et al., 2025a) and BAT (Long et al., 2023) use fixed mixing ratios, our approach maps erank to a linear weighting function, assigning a larger proportion of attention-based tokens for low-erank (simple) images and a larger proportion of diversity-based to-

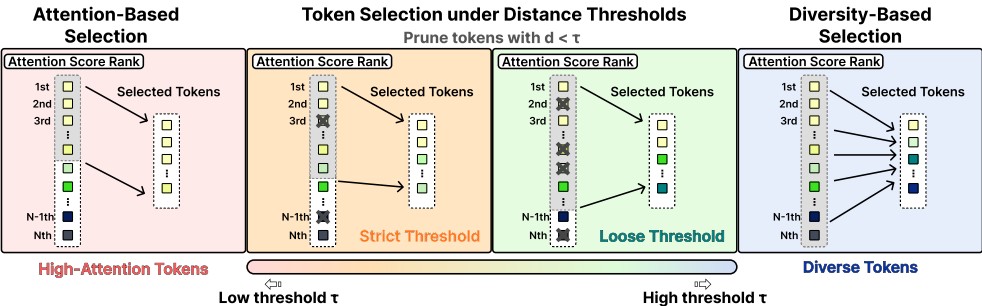

Figure 4: **Effect of similarity threshold $\tau$ on token selection.** A low (*strict*) $\tau$ prioritizes high-attention tokens, while a high (*loose*) $\tau$ increases the diversity of the selected tokens.

kens for high-erank (complex) images. Applying this adaptive rule to VisPruner and BAT leads to consistent accuracy improvements across both 128 and 64 token settings, as shown in Table 5. Conversely, an inverse adaptation—which deliberately assigns more importance tokens to high-erank images—results in a clear performance drop across benchmarks. Furthermore, as shown in Table 6, we also applied the same adaptive rule to a pairwise combination of a purely attention-based method and a purely diversity-based method, FasterVLM + DivPrune. Even in this setting, the adaptive rule achieves higher accuracy than a fixed, non-adaptive combination.

**Empirically guided adaptive similarity thresholding.** We further introduce a simple threshold-based pruning procedure that operationalizes the empirical behaviors identified above. Our method iteratively selects high-attention tokens and prunes similar neighbors, thereby modulating the diversity of the final token set based on the chosen threshold. The process is as follows:

1. All tokens are sorted in descending order based on their attention score.

2. Starting with the highest ranked token, we select it and then prune all other candidate tokens whose cosine distance $d$ to the selected token is smaller than an adaptive threshold $\tau_i$.

3. The process moves to the next highest-ranked token that has not been pruned and repeats the pruning step until the desired number of tokens is selected.

In this framework, the threshold $\tau$ is the parameter that directly governs the diversity of the final token set. As illustrated Fig. 4, applying a low (strict) threshold results in the pruning of only a few, highly similar tokens. Conversely, a high (loose) threshold removes a wider range of similar tokens, constructing a final token set with greater diversity. Detailed quantitative experiments verifying this correlation between $\tau$ and token diversity (measured via erank) and performance under different image complexity levels are provided in Appendix B.4.

However, the optimal level of diversity is dictated by the image's internal characteristics, leading to contrasting outcomes. For images with low erank & attention entropy(i.e., concentrated information), where critical information is focused in high-attention tokens, even highly similar tokens can contain vital fine-grained details. Consequently, aggressive pruning with a high threshold ($\tau$) degraded performance, whereas a conservative low threshold proved more effective. In contrast, for images with high erank & attention entropy (i.e., dispersed information), where information is more distributed and redundant, a higher threshold improved performance by effectively eliminating this redundancy and facilitating the selection of a more diverse token set.

Building on these empirical insights, we formulate a statistics-driven adaptive strategy that intrinsically adjusts to image complexity. To directly translate our findings into a robust mechanism, we define the dynamic threshold $\tau_i$ based on the normalized image complexity:

$$\tau_i = \text{order}_i \times \left( \frac{\text{erank}_{\text{input}}}{\text{erank}_{\text{avg}}} \times 0.01 \right) \tag{6}$$

where $\text{order}_i$ represents the rank of the token (1-based), and $\text{erank}_{\text{avg}}$ is the average effective rank of the LLaVA training set. To ensure stability, the final threshold is capped by a statistical upper bound $\tau_{max}$. This formulation naturally embodies our findings through the following mechanism:

| Method | VQA$^{v2}$ | GQA | VizWiz | SQA$^{IMG}$ | TextVQA | POPE | MME | MMB | MMB$^{CN}$ | Average |
|---|---|---|---|---|---|---|---|---|---|---|
| *Vanila 576 Tokens* | | | | | | | | | | |
| LLaVA-1.5-7B | 78.5 | 61.9 | 50.1 | 69.5 | 58.2 | 85.9 | 1862 | 64.7 | 58.1 | 100.00% |
| *Retain 128 Tokens* | | | | | | | | | | |
| FastV (ECCV'24) | 71.0 | 54.0 | 51.9 | 69.2 | 56.4 | 68.2 | 1490 | 63.0 | 55.9 | 92.31% |
| PDrop (CVPR'25) | 74.3 | 57.1 | 49.4 | 70.1 | 56.7 | 77.5 | 1696 | 62.3 | 55.3 | 95.17% |
| SparseVLM (ICML'25) | 75.1 | 57.3 | 49.7 | 69.0 | 56.3 | 83.1 | 1761 | 62.6 | 56.9 | 96.61% |
| PruMerge+ (ICCV'25) | 75.0 | 58.2 | 53.7 | 69.1 | 54.0 | 83.1 | 1554 | 61.8 | 55.8 | 95.64% |
| VisionZip (CVPR'25) | 75.6 | 57.6 | 51.6 | 68.7 | 56.9 | 83.3 | 1763 | 62.1 | 57.0 | 97.19% |
| VisPruner (ICCV'25) | 75.8 | 58.2 | 52.7 | 69.0 | 57.0 | 84.6 | 1768 | 62.7 | 57.3 | 98.01% |
| DivPrune (CVPR'25) | 76.0 | 59.4 | 52.8 | 68.6 | 54.5 | 85.5 | 1707 | 60.1 | 52.3 | 97.25% |
| **Ours** | 76.4 | 59.4 | 53.0 | 68.6 | 57.0 | 87.4 | 1748 | 61.8 | 55.5 | **98.04%** |
| *Retain 64 Tokens* | | | | | | | | | | |
| FastV (ECCV'24) | 55.9 | 46.0 | 49.1 | 70.1 | 51.6 | 35.5 | 1256 | 50.1 | 42.1 | 76.86% |
| PDrop (CVPR'25) | 56.3 | 46.1 | 46.3 | 68.8 | 49.2 | 40.8 | 1505 | 48.0 | 36.6 | 76.41% |
| SparseVLM (ICML'25) | 66.9 | 52.0 | 49.4 | 69.2 | 52.1 | 69.7 | 1561 | 58.3 | 49.6 | 88.60% |
| PruMerge+ (ICCV'25) | 71.3 | 55.4 | 53.7 | 69.5 | 52.0 | 75.7 | 1640 | 59.6 | 52.1 | 92.76% |
| VisionZip (CVPR'25) | 72.4 | 55.1 | 52.9 | 68.7 | 55.5 | 77.0 | 1690 | 60.1 | 55.4 | 94.46% |
| VisPruner (ICCV'25) | 72.7 | 55.4 | 53.3 | 69.1 | 55.8 | 80.4 | 1650 | 61.3 | 55.1 | 95.07% |
| DivPrune (CVPR'25) | 74.1 | 57.5 | 53.6 | 68.0 | 54.5 | 85.5 | 1615 | 60.1 | 52.3 | 95.02% |
| **Ours** | 75.5 | 57.4 | 54.0 | 68.6 | 56.0 | 84.1 | 1703 | 60.7 | 55.8 | **96.76%** |
| *Retain 32 Tokens* | | | | | | | | | | |
| PruMerge+ (ICCV'25) | 65.6 | 52.9 | 53.5 | 67.9 | 49.2 | 66.7 | 1550 | 55.1 | 45.9 | 87.01% |
| VisionZip (CVPR'25) | 67.1 | 51.8 | 52.4 | 69.1 | 53.1 | 69.4 | 1579 | 57.0 | 50.3 | 89.41% |
| Vispruner (ICCV'25) | 67.7 | 52.2 | 53.0 | 69.2 | 53.9 | 72.7 | 1538 | 58.4 | 52.7 | 90.75% |
| DivPrune (CVPR'25) | 71.2 | 54.9 | 53.3 | 68.6 | 52.9 | 81.5 | 1594 | 57.6 | 49.1 | 92.16% |
| **Ours** | 74.0 | 54.1 | 53.4 | 69.0 | 54.5 | 80.1 | 1603 | 60.4 | 53.6 | **94.02%** |

Table 7: **Results of different token pruning methods on 9 multimodal benchmarks.** *Average* is normalized to the full-token **LLaVA-1.5-7B** (set to 100%). MME is reported in its original score units.

- **Complex images** (erank$_{input}$ > erank$_{avg}$): A larger scaling factor increases $\tau_i$, enabling stronger pruning and promoting token diversity.
- **Simple images** (erank$_{input}$ < erank$_{avg}$): A smaller factor keeps $\tau_i$ low, preserving fine-grained, high-attention tokens.

This formulation ensures consistent adaptation across varying image complexities by grounding the threshold in dataset statistics.

## 5 EXPERIMENTS

**Baselines and Models** We apply our method to the widely adopted open-source model LLaVA-1.5-7B (Liu et al., 2024b), which is a fine-tuned variant of the LLaMA family. Based on this architecture, we compare our approach with several vision token pruning techniques. The baselines include methods leveraging attention scores within the LLM (FastV (Chen et al., 2024), SparseVLM (Zhang et al., 2025b), PyramidDrop (Xing et al., 2024)), approaches utilizing attention scores from the vision encoder (VisionZip (Yang et al., 2025), VisPruner (Zhang et al., 2025a)), and Diversity-based strategies such as DivPrune (Alvar et al., 2025). To ensure consistent, fair, and reproducible evaluation, we fixed the pretrained weights of LLaVA-1.5-7B and set the temperature to 0 across all experiments to produce deterministic outputs.

**Datasets** We conduct evaluations on a total of nine multimodal benchmarks. VQAv2 (Goyal et al., 2017) and GQA (Hudson & Manning, 2019) are large-scale visual question answering datasets that assess general vision-language understanding. VizWiz (Gurari et al., 2018) and TextVQA (Singh et al., 2019) introduce more challenging scenarios involving accessibility-related queries and text recognition in images. ScienceQA (Lu et al., 2022) requires scientific knowledge for complex reasoning tasks. MME (Yin et al., 2024) provides a comprehensive metric for fine-grained multimodal understanding. Finally, MMBench and MMBench-CN (Liu et al., 2024c) serve as multilingual benchmarks that evaluate overall performance across diverse tasks and languages. In addition, as introduced in Section 4.2, we further analyze the hallucination problem by using the CHAIR dataset (Rohrbach et al., 2018) to quantitatively evaluate the occurrence of object hallucination in the model.

**Main results** We evaluate our adaptive thresholding approach on LLaVA-1.5, focusing on the effect of dynamic threshold adjustment on token diversity and downstream task performance. As

shown in Table 5, our method consistently preserves accuracy under aggressive pruning. With 128 tokens, our method achieves competitive performance, showing modest gains of 0.85% over VisionZip and 0.79% over DivPrune. When reduced to 64 tokens, attention-based pruning methods suffer more than 25% degradation, whereas our method incurs only a 3.24% drop and achieves a slight performance edge over VisionZip and DivPrune by 2.2% and 1.74%, respectively. While recent hybrids such as VisPruner (Zhang et al., 2025a), are primarily attention-based approaches that modestly complement their selection with distant tokens for diversity, they remain non-adaptive and thus lack robustness, particularly on datasets where attention-based pruning is weak, such as POPE and MME datasets. The efficiency analysis is provided in Appendix A, and additional results on a broader set of LVLMs—including Qwen2.5-VL-7B, LLaVA-1.5-13B, and LLaVA-NeXT-7B—are reported in Appendix B.1.

In summary, our empirical analysis reveals the distinct tendencies of existing pruning approaches and demonstrates that an adaptive approach is essential to balance information preservation and diversity. Building on these findings, we establish that adaptive thresholding provides a principled and effective alternative to fixed or non-adaptive methods.

**Hallucination analysis.** As shown in Table 8 and consistent with the observations in Section 4.2, our empirical analysis on the CHAIR dataset reveals clear contrasts between pruning strategies. Diversity-based methods generally achieve higher hallucination scores ($C_S$, $C_I$) with higher recall, whereas attention-based methods show the opposite trend. This trade-off is likely because tokens with high attention scores tend to contain more reliable visual information, which is essential for mitigating hallucination.

| Method | Retain 64 Tokens | | | | Retain 128 Tokens | | | |
|---|---|---|---|---|---|---|---|---|
| | $C_s\downarrow$ | $C_i\downarrow$ | Recall$\uparrow$ | Len | $C_s\downarrow$ | $C_i\downarrow$ | Recall$\uparrow$ | Len |
| LLaVA-1.5-7B | 51.0 | 13.9 | 78.7 | 101.4 | 51.0 | 13.9 | 78.7 | 101.4 |
| *Attention-based methods* | | | | | | | | |
| FasterVLM (arXiv'24) | 45.4 | 13.5 | 69.3 | 94.0 | 45.8 | 13.3 | 75.4 | 97.0 |
| PruMerge+ (ICCV'25) | 45.2 | 15.6 | 66.7 | 91.4 | 46.8 | 14.4 | 71.5 | 95.2 |
| Vispruner (ICCV'25) | 49.8 | 15.0 | 72.6 | 96.7 | 52.8 | 15.4 | 77.1 | 98.7 |
| *Diversity-based methods* | | | | | | | | |
| DivPrune (CVPR'25) | 57.4 | 18.0 | 76.4 | 101.1 | 58.6 | 18.1 | 78.4 | 103.1 |
| FPSPruner | 58.6 | 18.6 | 76.0 | 100.5 | 59.4 | 18.8 | 81.1 | 104.1 |
| **Ours** | **52.2** | **15.9** | **75.7** | **99.1** | **54.4** | **16.5** | **78.1** | **101.1** |

Table 8: CHAIR evaluation (64 /128 tokens).

Further analysis indicates that methods primarily relying on token diversity tend to show weaker performance on overall benchmarks, as they do not incorporate attention scores and are less effective in capturing concentrated and reliable information. Conversely, attention-based methods preserve such concentrated tokens but lack diversity, which restricts their ability to handle questions involving multiple objects.

Building on these empirical findings, we propose an adaptive method that balances attention and diversity. This approach achieves 52.2 on $C_S$, 15.9 on $C_I$, and a recall of 75.7, which are close to the values obtained with the full set of visual tokens. These results emphasize findings from empirical analysis in clarifying the tendencies of different pruning strategies and provide evidence that adaptive approaches are needed to better align with varying image characteristics.

## 6 CONCLUSION

In this work, we presented a systematic empirical study of visual token pruning in LVLMs. Our analysis using effective rank and attention entropy revealed two consistent behavioral patterns that have not been previously characterized: (i) pruning methods differ substantially in the diversity they retain, and this retained diversity is closely linked to hallucination tendencies; and (ii) the effectiveness of attention-based versus diversity-based pruning shifts predictably with image complexity, with simple images favoring attention-based selection and complex images benefiting from diversity-based retention. These findings provide a unified perspective that explains when and why different pruning strategies succeed or fail. Building on this empirical understanding, we showed that incorporating image-aware adjustments into existing hybrid and mixed pruning methods yields consistent improvements across benchmarks, demonstrating that the empirical principles we identify are broadly applicable and model-agnostic. We further provided a minimal instantiation of these principles through an adaptive thresholding procedure, which achieves strong performance and reduces hallucination while significantly lowering computational cost. Overall, our study highlights the importance of understanding the underlying behaviors of pruning paradigms in LVLMs and offers empirical principles that can guide the design of future adaptive pruning strategies.

ACKNOWLEDGMENTS

This research was supported by grants from the National Research Foundation of Korea(NRF) funded by the Ministry of Education (No. RS-2025-25433078) and the Korea government(MSIT) (No. RS-2024-00456152). This work was also supported by LG Electronics, and the High-Performance Computing Support NPU Program of the National IT Industry Promotion Agency(NIPA) (No. N2025-0158). Computational resources were provided by the Cluster Server for Computational Science at Pusan National University.

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

APPENDIX TABLE OF CONTENTS

| Method | Retain Tokens | FLOPs (T) | Latency (ms/sample) | GPU Memory (GB) | Accuracy |
|---|---|---|---|---|---|
| Vanilla (LLaVA-1.5-7B) | 576 | 3.14 | 172 | 13.60 | 58.2 |
| PDrop (CVPR'25) | 64 | 0.51 | 128 | 13.30 | 55.0 |
| SparseVLM (ICML'25) | 64 | 0.52 | 129 | 16.26 | 55.2 |
| DivPrune (CVPR'25) | 64 | 0.48 | 110 | 13.30 | 55.8 |
| VisPruner (ICCV'25) | 64 | 0.48 | 115 | 13.30 | 55.4 |
| **Ours** | 64 | 0.48 | 115 | 13.30 | **56.0** |

Table 9: Efficiency and accuracy comparison **on single RTX 4090 at TextVQA dataset.** All models are evaluated under identical settings.

| Batch Size | 1 | 3 | 5 | 10 |
|---|---|---|---|---|
| erank overhead (ms) | 3.65 | 10.21 | 16.84 | 33.40 |

Table 10: erank computation time across different batch sizes **on single RTX 4090.**

## A  EFFICIENCY ANALYSIS

**Computation Overhead of Attention entropy and erank.**   The erank quantifies the representational complexity of a feature matrix $X \in \mathbb{R}^{N \times D}$ based on the distribution of its singular spectrum. To improve computational efficiency, we compute the covariance matrix across tokens using a fast $N \times N$ formulation. The definition is given as follows:

$$C = XX^\top, \quad S = \sqrt{\lambda(C)}, \quad p_i = \frac{S_i}{\sum_j S_j}, \quad \mathrm{erank}(X) = \exp\left(-\sum_i p_i \log p_i\right). \quad (7)$$

Here, $C$ denotes the $N \times N$ covariance matrix, $\lambda(C)$ its eigenvalue spectrum, $S$ the square roots of these eigenvalues (i.e., singular values), and $p_i$ the normalized spectral ratio. Thus, the erank corresponds to the exponential of the Shannon entropy of the normalized spectrum. Eq. (7) is equivalent to the SVD-based definition, since the singular values of $X$ match the square roots of the eigenvalues of $XX^\top$.

In typical LVLM settings, the number of tokens is much smaller than the feature dimension ($N \ll D$). For example, LLaVA-7B-1.5 uses $N=576$ visual tokens with a $D=4096$ embedding. This allows computing erank efficiently using the smaller covariance matrix $C \in \mathbb{R}^{N \times N}$.

To assess the practical cost of these metrics, we compare their theoretical complexity with measured runtime. Naively applying SVD to $X \in \mathbb{R}^{N \times D}$ has $\mathcal{O}(ND^2)$ complexity, since decomposition is performed on a tall $N \times D$ matrix. In contrast, the fast formulation computes the spectrum of the smaller covariance matrix $C = XX^\top$, reducing the overall cost to $\mathcal{O}(N^2D + N^3)$ when $N \ll D$.

We further measure the per-image runtime under the LLaVA-1.5-7B configuration. Computing erank takes 3.4 ms on average, while the full model inference takes 115 ms. Thus, **erank accounts for only ~3.2% of the total inference time, remaining lightweight in practice**. During batched inference, As shown in Table 10, the cost scales approximately linearly with the number of images, as both covariance construction and eigenvalue computation operate independently across samples.

**Efficiency Comparison.**   As shown in Table 9, the proposed method reduces FLOPs by **89%** under the 64-token setting, while still preserving **96.2%** of the original performance compared to the vanilla LLaVA-1.5-7B model. Notably, our method outperforms in-LLM pruning approaches such as SparseVLM (Zhang et al., 2025b)and PyramidDrop (Xing et al., 2024) in terms of accuracy, achieving a better efficiency–performance trade-off. Meanwhile, when compared with recent pre-pruning approaches such as VisPruner (Zhang et al., 2025a) and DivPrune (Alvar et al., 2025), the computational indicators (FLOPs, latency, GPU memory) remain nearly identical, while our method still attains the highest accuracy among them.

| Method | GQA | SQA$^{IMG}$ | POPE | MME | TextVQA | Rel. |
|---|---|---|---|---|---|---|
| *All 576 Tokens* | | | | | | |
| LLaVA-1.5-7B | 61.9 | 69.5 | 85.9 | 1862 | 58.2 | 100.0% |
| *Retain 128 Tokens* | | | | | | |
| Erank-based | 59.4 | 68.6 | 87.4 | 1748 | 57.0 | 98.13% |
| Att. entropy-based | 59.4 | 68.6 | 87.0 | 1721 | 56.9 | 98.06 % |
| *Retain 64 Tokens* | | | | | | |
| Erank-based | 57.4 | 68.6 | 84.1 | 1703 | 56.0 | 95.97% |
| Att. entropy-based | 57.7 | 68.7 | 83.2 | 1690 | 56.0 | 95.84% |

Table 11: erank-based vs. entropy-based thresholding.

These three methods—VisPruner, DivPrune, and ours— share the property of performing pre-pruning before the LLM input, which substantially reduces the internal computation of the LLM. Compared to pruning inside intermediate layers of the LLM, pre-pruning provides a much stronger efficiency gain since the token reduction applies to all subsequent layers, yielding significant savings in FLOPs, memory, and latency with minimal overhead. In contrast, pruning at intermediate layers inside the LLM, as exemplified by methods such as SparseVLM and PyramodDrop, allows richer contextualization before tokens are removed and thus carries a lower risk of discarding important information, but its efficiency benefit is limited because the early layers still process the full set of tokens. Therefore, pre-pruning is preferable in terms of efficiency.

In addition, our method is fully compatible with FlashAttention (Dao et al., 2022), enabling further efficiency gains when combined with state-of-the-art acceleration techniques. Overall, these results demonstrate that our method strikes an effective balance between computational efficiency and accuracy.

# B  ADDITIONAL RESULTS

## B.1  EVALUATION ON OTHERS MODEL

In addition to LLaVA-1.5-7B, we also conducted experiments on larger and architecturally diverse LVLMs, including LLaVA-1.5-13B (576 tokens), LLaVA-NeXT-7B (2880 tokens), and Qwen2.5-VL-7B. Across all these settings, our method consistently demonstrated stable and strong performance, further validating the effectiveness and generality of our approach. Detailed results are presented in Table 14, Table 15, and Table 16.

## B.2  ENTROPY-BASED ADAPTIVE THRESHOLDING

We first examined the relationship between attention entropy and effective rank (erank). As shown in Figure 5, the two metrics exhibit a moderate Pearson correlation of 0.63 across the full MME dataset, indicating that both capture a related notion of visual information dispersion. While entropy reflects the spread of attention weights, erank additionally incorporates feature-space geometry through its singular-value spectrum.

Motivated by this correlation, we investigated whether entropy can directly replace erank in our adaptive thresholding rule. Specifically, we substituted attention entropy for erank in the formulation of Eq. 6 and computed the reference average entropy once using the LLaVA training set.

Table 11 presents results across multiple benchmarks at token budgets of 64 and 128. Entropy-

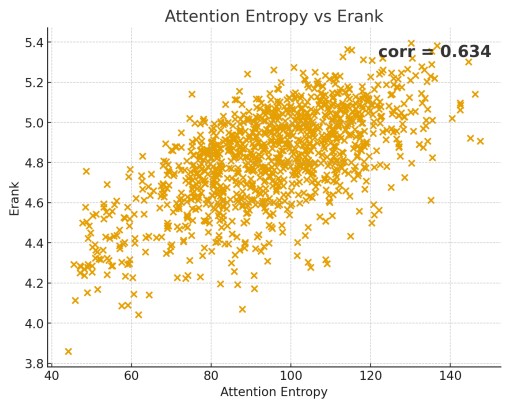

Figure 5: Attention entropy vs. erank on MME dataset.

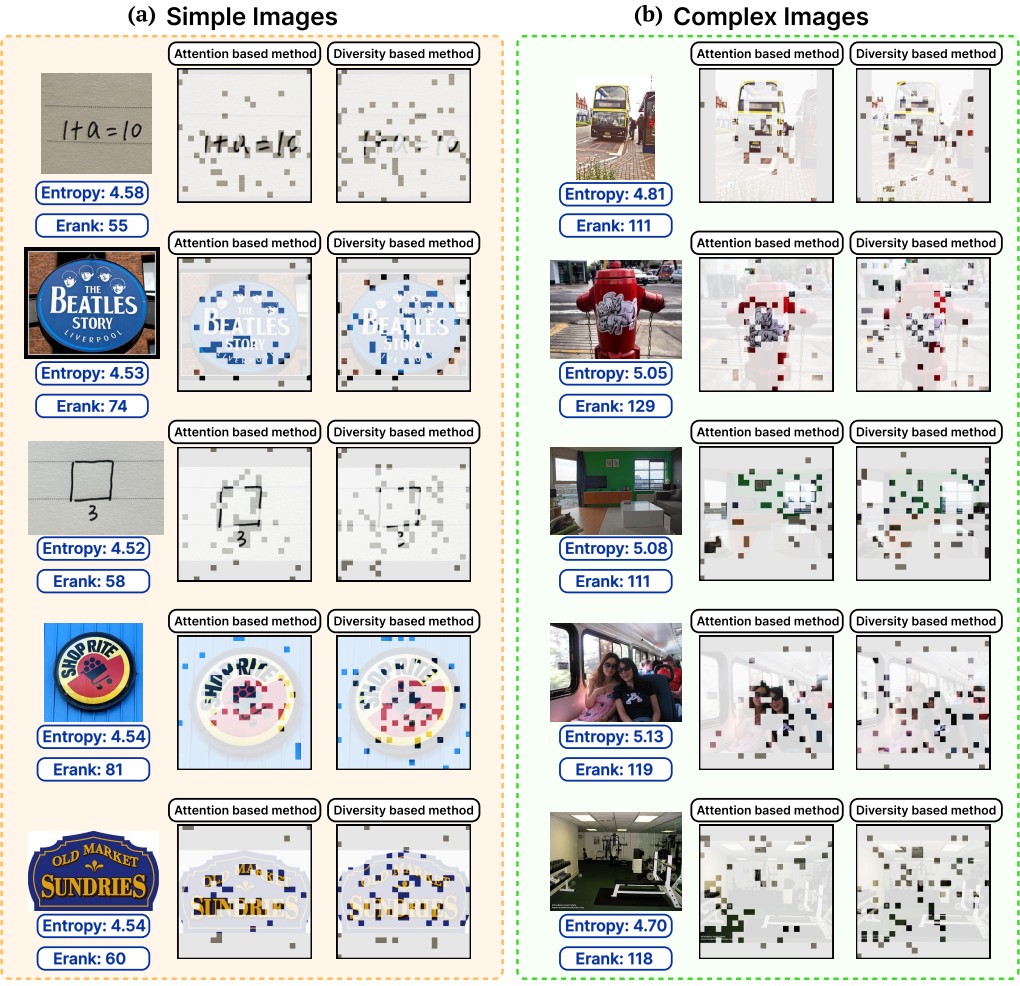

Figure 6: **Extended examples for simple vs. complex images.** The same trend as in Fig. 3 is observed: attention-based methods work well on simple images, while diversity-based methods cover complex images more broadly.

based adaptation yields performance trends closely aligned with those of erank-based adaptation, with only minor variations about 0.13.

### B.3 SUPPLEMENTARY EXAMPLES OF IMAGE COMPLEXITY–DEPENDENT PRUNING DIFFERENCES

As shown in Fig. 6, the qualitative patterns observed consistently reproduced across additional samples. For simple images (with low entropy and erank), attention-based pruning effectively captures the concentrated regions, while the additional benefits of diversity-based pruning are limited. Conversely, for complex images (with higher entropy and erank), diversity-based pruning ensures broader coverage, highlighting its strength in dispersed scenarios. These supplementary examples reinforce that image complexity is a key determinant of pruning effectiveness and motivate the need for an adaptive strategy that integrates both approaches.

### B.4 EFFECT OF SIMILARITY THRESHOLD ON TOKEN DIVERSITY

We varied $\tau$ from 0 to 0.25 and observed its impact on token diversity. As shown in Table 12, the results demonstrate a direct positive correlation between the similarity threshold $\tau$ and the diversity

| Metric | Similarity Threshold ($\tau$) | | | | | | |
|---|---|---|---|---|---|---|---|
| | **0** | **0.01** | **0.05** | **0.1** | **0.15** | **0.20** | **0.25** |
| *Erank / Performance* | | | | | | | |
| *High erank (complex images) datasets* | | | | | | | |
| **MME (High)** | 13.85 / 1362 | 16.72 / 1358 | 17.44 / 1368 | 18.58 / 1379 | 19.55 / 1363 | 20.13 / **1381** | 19.60 / 1380 |
| **POPE** | 14.40 / 77.5 | 17.13 / 77.7 | 17.81 / 79.1 | 18.94 / 81.5 | 19.83 / 83.4 | 20.43 / 85.3 | 20.49 / **86.1** |
| *Low erank (simple images) datasets* | | | | | | | |
| **MME (Low)** | 11.32 / 309 | 14.46 / **311** | 15.38 / 307 | 16.39 / 292 | 17.21 / 304 | 17.00 / 284 | 15.50 / 290 |
| **ScienceQA** | 11.91 / **69.1** | 15.29 / 68.9 | 16.22 / 68.9 | 17.33 / 68.2 | 18.09 / 68.3 | 18.45 / 68.7 | 16.50 / 68.5 |

Table 12: Comparison of performance across different similarity thresholds. For each metric, the boldface indicates the best performance.

| Method | Avg. tokens | GQA | SQA$^{IMG}$ | POPE | MME | MMBench | Rel. |
|---|---|---|---|---|---|---|---|
| LLaVA-1.5-7B | 576 | 61.9 | 69.5 | 85.9 | 1510 | 64.3 | 100.0% |
| ATP-LLaVA (CVPR'25) | 88 | 56.8 | 67.2 | 82.8 | 1401 | 64.7 | 95.1% |
| Our (fixed count) | 88 | 58.1 | 68.0 | 84.2 | 1405 | 62.3 | 95.4 % |
| **Our (Adaptive count)** | 85.5 | 58.4 | 68.2 | 85.2 | 1408 | 63.1 | **96.0%** |

Table 13: Results of adaptive-count vs. fixed-count pruning

of the selected token set. As $\tau$ increases, the erank of selected tokens consistently rises across all datasets. This indicates that a higher threshold causes more tokens to be treated as redundant and pruned, which in turn enhances the diversity of the final set. Notably, token diversity was at its lowest when the threshold was close to zero, as very little similarity-based pruning occurs under this condition.

While increasing $\tau$ consistently improves token diversity, its impact on performance varies depending on image complexity. *For more complex images with higher erank*, a larger $\tau$ helps eliminate redundant tokens and improves performance by encouraging a more diverse token representation. In contrast, *for images with lower complexity*, where informative content is concentrated in a small number of high-attention tokens, overly aggressive pruning with a large $\tau$ can remove fine-grained but important details, leading to degraded performance.

### B.5 EXTENDING OUR ANALYSIS TO ADAPTIVE TOKEN COUNT

Recent approaches—including ATP-LLaVA (Ye et al., 2025)—have actively explored adaptive token count strategies that adjust the number of retained visual tokens per input. Our empirical analysis naturally extends to this direction. High-erank (complex) images contain more dispersed visual information and therefore benefit from retaining a more diverse set of tokens, whereas low-erank (simple) images have more concentrated information and are better represented by selecting a smaller set of focused tokens. Based on this analysis, we design an adaptive-count strategy that preserves more tokens for complex images and prunes more aggressively for simple ones. For comparability with ATP-LLaVA, we adopt the same reference budget of 88 tokens, reducing the count by up to 20% for low-erank images and increasing it proportionally for high-erank images. Since the number of retained tokens varies across inputs, we report dataset-level averages. As shown in Table 13, the adaptive-count variant—retaining an average of approximately 85.5 tokens—achieves higher performance than both ATP-LLaVA and our fixed-count baseline on GQA, SQA, POPE, MME, and MMBench, while also providing slight efficiency gains. These results confirm that adjusting the token count aligns naturally with our erank-based analysis and enables more effective allocation of computational budget according to image complexity.

| Method | VQA$^{v2}$ | GQA | VizWiz | SQA$^{IMG}$ | TextVQA | POPE | MME | MMB | MMB$^{CN}$ | Average |
|---|---|---|---|---|---|---|---|---|---|---|
| *Vanilla 576 Tokens* | | | | | | | | | | |
| LLaVA-1.5-13B | 80.0 | 63.3 | 53.6 | 72.8 | 61.2 | 86.0 | 1531 | 68.5 | 63.5 | 100% |
| *Retain 128 Tokens* | | | | | | | | | | |
| FastV (ECCV'24) | 75.3 | 58.3 | 54.6 | 74.2 | 58.6 | 75.5 | 1460 | 66.1 | 62.3 | 96.0% |
| PDrop (CVPR'25) | 78.2 | 61.0 | 53.8 | 73.3 | 60.2 | 83.6 | 1489 | 67.5 | 62.8 | 98.4% |
| SparseVLM (ICML'25) | 77.6 | 59.6 | 51.4 | 74.3 | 59.3 | 85.0 | 1488 | 68.4 | 62.6 | 97.8% |
| PruMerge+ (ICCV'25) | 76.2 | 58.3 | 52.8 | 73.3 | 56.1 | 82.7 | 1446 | 66.3 | 61.2 | 95.8% |
| VisionZip (CVPR'25) | 76.8 | 57.9 | 52.3 | 73.8 | 58.9 | 82.7 | 1450 | 67.4 | 62.5 | 96.7% |
| DivPrune (CVPR'25) | 77.1 | 59.2 | 53.5 | 72.8 | 58.0 | 86.8 | 1458 | 66.3 | 60.7 | 97.0% |
| Ours | 77.5 | 59.1 | 52.5 | 72.8 | 58.9 | 86.9 | 1481 | 67.6 | 61.9 | **97.6%** |
| *Retain 64 Tokens* | | | | | | | | | | |
| FastV (ECCV'24) | 65.3 | 51.9 | 53.8 | 73.1 | 53.4 | 56.9 | 1246 | 59.2 | 55.1 | 85.8% |
| PDrop (CVPR'25) | 70.8 | 54.1 | 50.5 | 73.1 | 55.3 | 66.1 | 1247 | 63.1 | 56.6 | 88.7% |
| SparseVLM (ICML'25) | 73.2 | 55.9 | 52.1 | 73.0 | 57.1 | 77.9 | 1374 | 65.2 | 60.3 | 93.5% |
| PruMerge+ (ICCV'25) | 72.6 | 56.3 | 52.4 | 73.5 | 54.4 | 75.7 | 1338 | 65.0 | 59.3 | 92.3% |
| VisionZip (CVPR'25) | 73.7 | 56.2 | 53.2 | 74.2 | 57.4 | 75.7 | 1380 | 64.9 | 61.3 | 93.9% |
| DivPrune (CVPR'25) | 75.2 | 57.9 | 54.4 | 71.7 | 57.4 | 84.5 | 1454 | 64.1 | 59.8 | 95.6% |
| Ours | 75.7 | 57.5 | 54.2 | 72.0 | 58.6 | 82.0 | 1437 | 66.2 | 61.6 | **96.0%** |

Table 14: **Results of different token pruning methods on 9 multimodal benchmarks.** *Average* is normalized to the full-token **LLaVA-1.5-13B** (set to 100%). MME is reported in its original score units, and it is included only in the *Perception* section to enable broader comparison with existing methods.

| Method | VQA$^{v2}$ | GQA | VizWiz | SQA$^{IMG}$ | TextVQA | POPE | MME | MMB | MMB$^{CN}$ | Average |
|---|---|---|---|---|---|---|---|---|---|---|
| *Vanilla 2880 Tokens (Upper Bound)* | | | | | | | | | | |
| LLaVA-NeXT-7B | 81.3 | 62.5 | 55.2 | 67.5 | 60.3 | 86.8 | 1512 | 65.8 | 57.3 | 100% |
| *Retain 640 Tokens* | | | | | | | | | | |
| FastV (ECCV'24) | 77.0 | 58.9 | 53.9 | 67.4 | 58.1 | 79.5 | 1412 | 63.1 | 53.5 | 95.2% |
| PDrop (CVPR'25) | 79.1 | 60.0 | 53.8 | 66.7 | 57.8 | 83.8 | 1475 | 64.1 | 55.2 | 97.0% |
| SparseVLM (ICML'25) | 79.2 | 61.2 | 53.6 | 67.6 | 59.7 | 85.3 | 1456 | 65.9 | 58.6 | 98.8% |
| PruMerge+ (ICCV'25) | 78.2 | 60.8 | 57.9 | 67.8 | 54.9 | 85.3 | 1480 | 64.6 | 57.3 | 98.3% |
| VisionZip (CVPR'25) | 79.1 | 61.2 | 57.1 | 68.1 | 59.9 | 86.0 | 1493 | 65.8 | 58.1 | 99.8% |
| DivPrune (CVPR'25) | 79.3 | 61.9 | 55.7 | 67.8 | 57.0 | 86.9 | 1469 | 65.8 | 57.3 | 98.9% |
| Ours | 79.3 | 62.0 | 56.0 | 67.8 | 59.0 | 86.1 | 1502 | 65.9 | 58.3 | **99.64%** |
| *Retain 320 Tokens* | | | | | | | | | | |
| FastV (ECCV'24) | 61.5 | 49.8 | 51.3 | 66.6 | 52.2 | 49.5 | 1099 | 53.4 | 42.5 | 79.9% |
| PDrop (CVPR'25) | 66.8 | 50.4 | 49.7 | 66.7 | 49.0 | 60.8 | 1171 | 55.5 | 44.7 | 82.5% |
| SparseVLM (ICML'25) | 74.6 | 57.9 | 54.2 | 67.2 | 56.5 | 76.9 | 1386 | 63.1 | 56.7 | 94.6% |
| PruMerge+ (ICCV'25) | 75.3 | 58.8 | 57.7 | 68.1 | 54.0 | 79.5 | 1444 | 63.0 | 55.6 | 95.7% |
| VisionZip (CVPR'25) | 76.2 | 58.9 | 56.2 | 67.5 | 58.8 | 82.3 | 1397 | 63.3 | 55.6 | 96.4% |
| DivPrune (CVPR'25) | 77.2 | 61.1 | 55.6 | 67.7 | 56.2 | 84.7 | 1423 | 63.9 | 55.7 | 97.0% |
| Ours | 77.8 | 60.1 | 55.8 | 67.3 | 58.4 | 84 | 1475 | 64.5 | 57.1 | **97.94%** |

Table 15: **Results of different token pruning methods on 9 multimodal benchmarks.** *Average* is normalized to the full-token **LLaVA-NeXT-7B** (set to 100%). MME is reported in its original score units, and it is included only in the *Perception* section to enable broader comparison with existing methods.

## C  FARTHEST POINT SAMPLING

Farthest Point Sampling (FPS) is one of the simplest methods that guarantees diversity. Starting from an initial point, it iteratively selects the point that is farthest from the already chosen set by measuring the distance to the nearest selected point. For each point $p_j$, the minimum distance to the selected set $S$ is defined as

$$d(p_j) = \min_{s \in S} \|p_j - s\|_2,$$

and the next point is chosen as

$$p_{i_t} = \arg \max_{p_j \in P \setminus S} d(p_j).$$

| Method | TextVQA | ChartQA | AI2D | OCRBench | MME (Perc.) | MME (Cogn.) | MMB-EN | MMB-CN | Rel. |
|---|---|---|---|---|---|---|---|---|---|
| *Vanilla 1225 Tokens* (100%) | | | | | | | | | |
| Qwen2.5-VL-7B | 82.1 | 77.5 | 83.0 | 84.1 | 1691 | 642 | 83.2 | 83.2 | 100.0% |
| *Retain 512 Tokens* (60.5% Reduction) | | | | | | | | | |
| DivPrune(CVPR25) | 79.7 | 72.0 | 81.9 | 78.5 | 1704 | 620 | 81.8 | 81.0 | 96.8% |
| **Ours** | 79.8 | 71.5 | 82.1 | 78.5 | 1700 | 630 | 81.3 | 80.8 | **96.9%** |
| *Retain 256 Tokens* (79.1% Reduction) | | | | | | | | | |
| DivPrune(CVPR25) | 75.2 | 62.9 | 79.7 | 66.4 | 1703 | 542 | 80.3 | 79.8 | 90.7% |
| **Ours** | 74.2 | 62.4 | 80.5 | 65.8 | 1670 | 630 | 80.9 | 80.3 | **92.1%** |

Table 16: **Performance comparison on Qwen2.5-VL-7B at different token retention ratios.** *Rel.* is the relative performance normalized to the full-token Qwen2.5-VL-7B (set to 100%). MME scores are reported as separate Perception (Perc.) and Cognition (Cogn.).

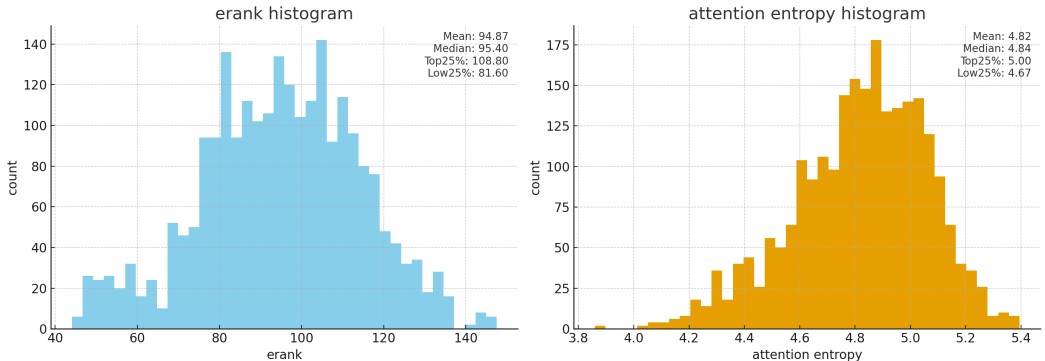

Figure 7: Histograms of erank and attention entropy measured on the MME dataset.

Repeating this process until the desired number $k$ is reached ensures that the selected points are evenly distributed across the data space, providing a more balanced representation than simple random sampling.

## D  ATTENTION ENTROPY AND ERANK SCALE

Entropy and erank exhibit fundamentally different scaling behaviors. As illustrated in the Figure 7, the entropy computed from CLIP-L (using 576 visual tokens) is concentrated within a very narrow range. Specifically, the mean entropy over all samples is 4.80, the median is 4.78, the upper quartile (Q3) is 4.96, and the lower quartile (Q1) is 4.63. In other words, entropy values predominantly vary within the interval of approximately 4.0–5.4. Therefore, even seemingly small differences (e.g., 0.3–0.5) represent meaningful relative changes within this restricted scale.

In contrast, erank spans a much broader scale. The mean erank is 94.87, the median is 95.40, the upper quartile is 108.80, and the lower quartile is 81.59. This wide spread demonstrates that erank captures coarser-grained variations in token correlation structure and spectral dispersion, complementing the fine-grained sensitivity of entropy.

## E  ROBUSTNESS ANALYSIS OF ERANK UNDER INPUT CORRUPTIONS

**Robustness of erank to Noisy or Degraded Inputs**  To evaluate the stability of the erank metric under degraded visual conditions, we conducted a sensitivity analysis following the corruption protocol of COCO-C (Hendrycks & Dietterich, 2019). Fifteen corruption types were applied to all 2,374 images in the MME dataset, and the resulting erank values were compared against those computed on clean images. Two corruption severities (1 and 3) were used (see Fig 8) corresponding to mild and moderate perturbation strengths commonly examined in robustness studies.

Table 17 summarizes the mean absolute deviation in erank and the corresponding relative change with respect to the clean-image mean. Overall, erank exhibited a high degree of stability across all

| Corruption Type | Severity = 1 | | Severity = 3 | |
|---|---|---|---|---|
| | Mean erank diff. | Relative change | Mean erank diff. | Relative change |
| **Zoom blur** | 5.85 | 6.14% | 7.62 | 8.12% |
| **Brightness** | 1.48 | 1.56% | 2.52 | 2.70% |
| **Contrast** | 2.38 | 2.54% | 4.38 | 4.62% |
| **Defocus blur** | 2.06 | 2.28% | 3.74 | 4.16% |
| **Elastic transform** | 2.18 | 2.54% | 3.38 | 4.05% |
| **Fog** | 3.24 | 3.73% | 4.05 | 4.71% |
| **Frost** | 3.96 | 4.56% | 6.10 | 7.24% |
| **Gaussian noise** | 2.83 | 3.23% | 4.17 | 4.73% |
| **Impulse noise** | 3.34 | 4.04% | 4.37 | 5.10% |
| **Jpeg compression** | 2.31 | 2.48% | 2.77 | 2.98% |
| **Motion blur** | 2.11 | 2.33% | 3.38 | 3.91% |
| **Pixelate** | 1.22 | 1.32% | 1.61 | 1.76% |
| **Shot noise** | 2.74 | 3.14% | 4.06 | 4.62% |
| **Snow** | 3.84 | 4.17% | 5.37 | 5.76% |
| **Glasses blur** | 2.15 | 2.46% | 4.07 | 4.79% |
| **Average** | **2.78** | **3.10%** | **4.11** | 4.62% |

Table 17: Sensitivity of erank under 15 corruption types from COCO-C, evaluated on the MME dataset.

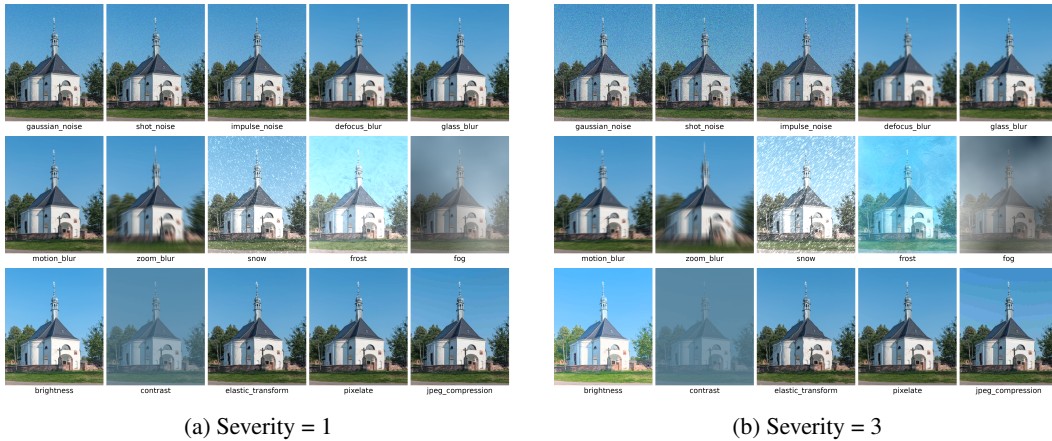

(a) Severity = 1            (b) Severity = 3

Figure 8: Visualization of the 15 COCO-C corruption types applied to MME images at severity levels 1 and 3.

corruption types. The average deviation was 2.78 at severity 1 and 4.11 at severity 3, representing only a small fraction of the natural variation in erank. For reference, the clean MME dataset spans a wide range of erank values (mean 94.86, min 44.08, max 147.49, standard deviation 19.46).

Corruptions that alter the *global spatial structure* of the image—such as zoom blur, frost, snow, and elastic transform—produced moderately larger deviations (typically 4–7 points), reflecting their broader impact on the singular-value spectrum. In contrast, distortions that modify only *local pixel-level appearance*, including brightness changes, pixelation, and JPEG compression, had minimal effect (1–2.5 points). Increasing corruption severity from 1 to 3 resulted in only marginal increases in deviation, indicating that erank maintains consistent behavior even under stronger degradation.

## F   QUALITATIVE ANALYSIS OF TOKEN SELECTION IN FINE-GRAINED REASONING TASKS

### F.1   TOKEN-SELECTION BEHAVIORS AND THEIR IMPACT ON FINE-GRAINED REASONING

The characteristics of the tokens selected by attention-based and diversity-based pruning influence not only hallucination generation but also the model's overall reasoning behavior. As shown in the examples in Fig. 9, the tendencies identified in Section 4.1 also appear in more fine-grained reasoning tasks such as counting and spatial relation inference. Accordingly, we conduct a qualitative analysis comparing how diversity-based, attention-based, and adaptive pruning methods select tokens and how these selections affect the open-ended responses they generate.

Examples 1 and 4 demonstrate that attention-based pruning, due to its narrow focus on specific regions, may overlook certain objects and produce incorrect answers. In contrast, diversity-based pruning preserves broader spatial information, captures more objects, and often provides more detailed descriptions, resulting in more accurate predictions in such cases. However, in Examples 2 and 5, the opposite pattern emerges: attention-based pruning correctly predicts the object count by concentrating on the primary instances, whereas diversity-based pruning—owing to its dispersed token selection—incorrectly infers the presence of additional objects, leading to hallucination. The adaptive method exhibits intermediate behavior between the two. Although it fails in Examples 1 and 4 by focusing on a narrower area than the diversity-based method, all three methods struggle with these particularly challenging cases. Conversely, in Examples 2 and 5, the adaptive method predicts the correct count alongside the attention-based method, while the diversity-based method hallucinates additional objects.

A similar pattern arises in spatial reasoning. As seen in Examples 3 and 6, the diversity-based method leverages a wider range of spatial cues and produces richer relational descriptions, but this increases the likelihood of inferring relations that do not actually exist. In contrast, the attention-based method provides stable, object-centered relational descriptions, though with more limited coverage compared to the diversity-based approach. The adaptive method again lies between the two, capturing the key objects while reducing the hallucinations introduced by the diversity-based method, ultimately yielding more stable spatial reasoning.

To further examine whether the adaptive method overlooks small or rare objects, we evaluated all three pruning strategies on an Existence task, which asks whether a specific object is present in the image (Fig. 10). In Example 1, the adaptive method successfully retained the necessary tokens to detect a small object that the attention-based method missed. Similarly, in Example 2, where both the attention-based and diversity-based methods failed to identify a very small object, the adaptive method preserved the relevant cues and produced the correct answer. These results indicate that the adaptive strategy reliably maintains important tokens and remains effective even when small objects serve as critical evidence for reasoning.

Overall, across diverse reasoning tasks, the adaptive method demonstrates robust and reliable behavior, as supported by both quantitative results and qualitative examples.

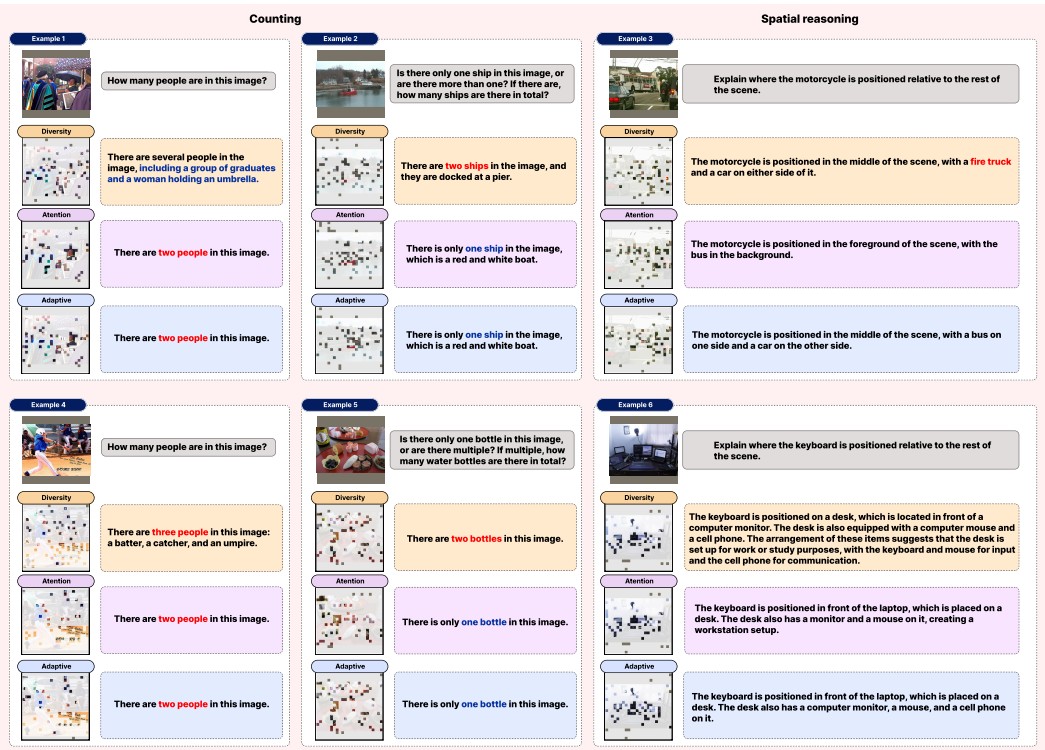

Figure 9: **Qualitative comparison of three pruning strategies—attention-based, diversity-based, and our adaptive method—on counting and spatial reasoning tasks.** Diversity-based pruning retains a broader range of spatial cues, often producing detailed descriptions but occasionally introducing hallucinated objects or relationships. Attention-based pruning focuses on salient regions, offering stable but sometimes overly narrow predictions. The adaptive method balances both mechanisms, mitigating hallucinations while preserving essential visual cues for reliable reasoning.

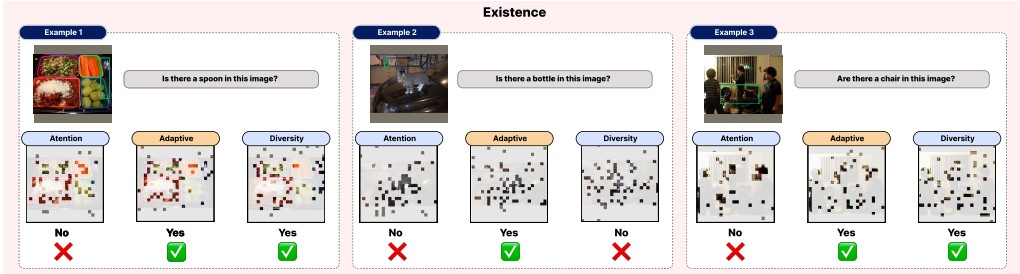

Figure 10: **Qualitative comparison on the Existence task.** Adaptive pruning correctly preserves tokens needed to identify small objects, outperforming attention- and diversity-based methods in challenging cases.

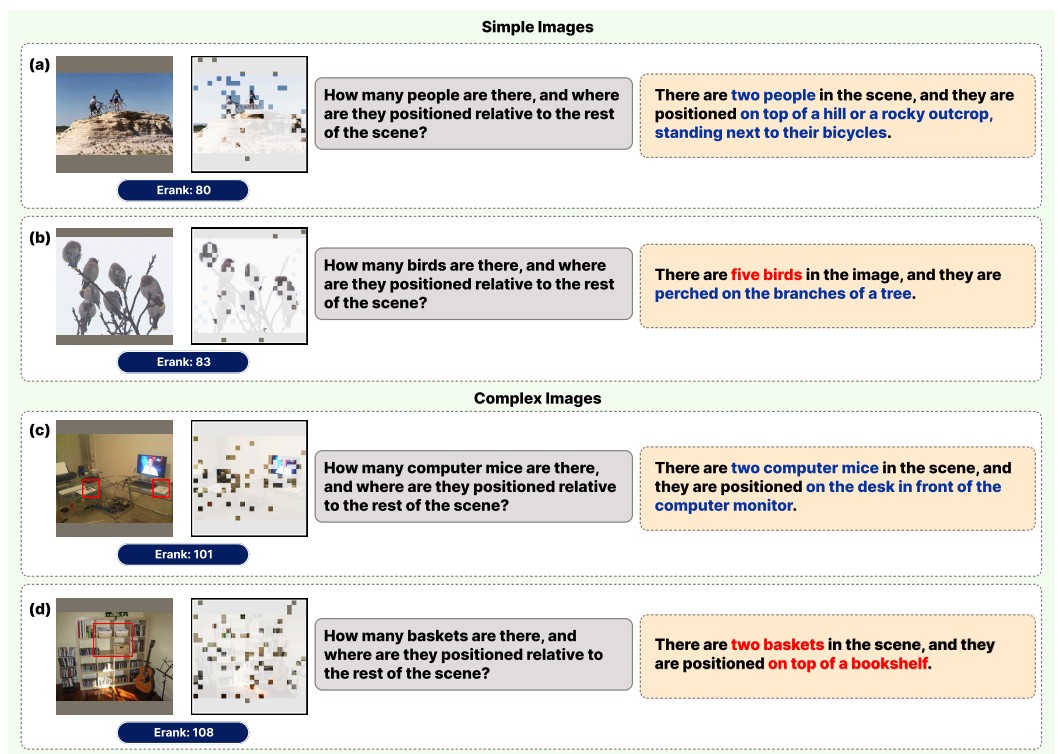

Figure 11: **Examples illustrating how image complexity governs the adaptive pruning method's success and failure in fine-grained reasoning.** For simple, low-erank images, the method performs well when key objects are spatially concentrated (a) but struggles when many similar objects are scattered across the scene (b). For complex, high-erank images, it succeeds when semantic cues are broadly distributed (c) but fails when crucial evidence for fine-grained reasoning is highly localized within a cluttered scene (d).

## F.2 FAILURE MODES OF THE ADAPTIVE METHOD IN FINE-GRAINED REASONING ACROSS IMAGE COMPLEXITY

We analyze cases in which the adaptive method—whose token selection is guided by image complexity—fails on fine-grained reasoning tasks. To do so, we use questions that jointly require object counting and spatial relation reasoning across images of varying complexity, and summarize the observed failure patterns below. Representative examples are shown in Fig. 11.

**(1) Low-erank images with many objects.** In simple images with low erank, the adaptive method allocates greater weight to high-attention tokens and compensates with limited diversity. This results in a selection pattern similar to attention-based pruning. When objects are few and localized (e.g., Fig. 11a), this focused selection enables accurate counting and spatial reasoning. However, when many objects are spread throughout the image (Fig. 11b), the concentrated selection fails to capture the broader spatial layout, leading to counting errors and incomplete reasoning.

**(2) High-erank images with locally concentrated key evidence.** For complex images with high erank, the adaptive method behaves more like diversity-based pruning and tends to select widely dispersed tokens. When objects or semantic cues are broadly distributed (Fig. 11c), this dispersed selection is advantageous and supports accurate counting and relational reasoning. However, when the crucial evidence is highly localized within the scene (Fig. 11d), the wide-spread token selection dilutes attention around the important region, causing the method to overlook key cues and fail in both counting and spatial inference.

.

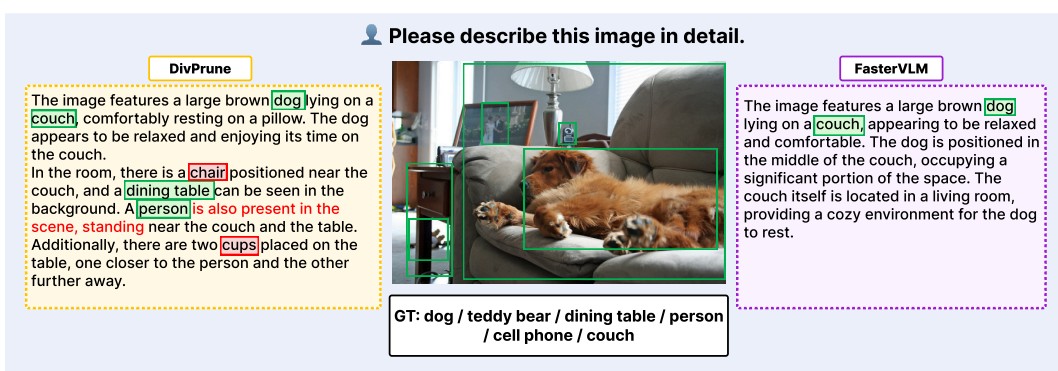

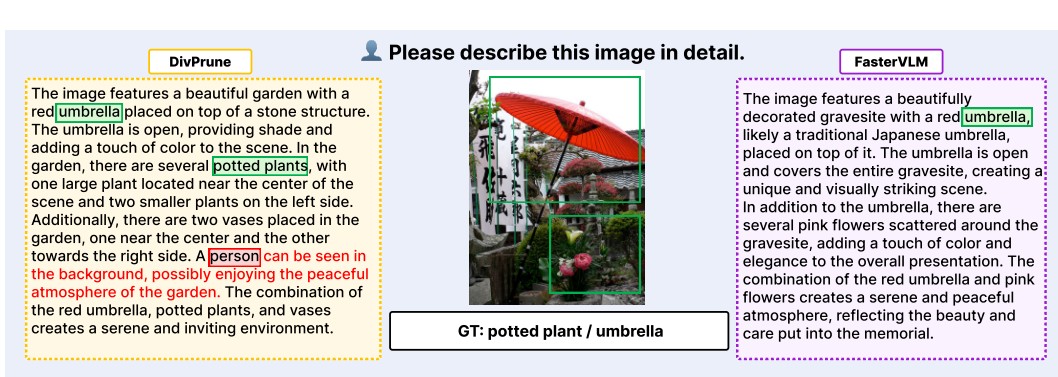

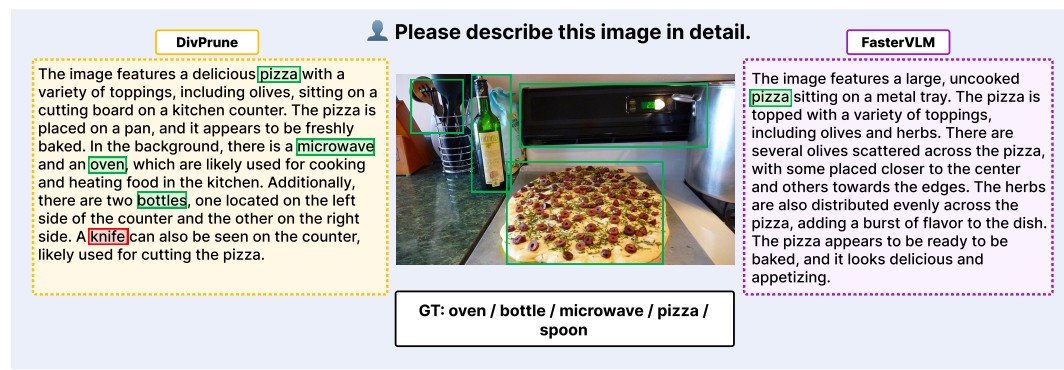

Figure 12: **CHAIR qualitative comparisons: FasterVLM vs. DivPrune (Set 1).** In the annotations, ■ GT Obj. and ■ Hallucinated Obj. label object words; **red text** indicates incorrect phrases.

## G  MORE EXAMPLES ON CHAIR

To further illustrate the differences between attention-based and diversity-based pruning in the image captioning task, we provide additional qualitative samples from the CHAIR dataset comparing FasterVLM as an attention-based method and DivPrune as a diversity-based method(Fig. 12 and Fig. 13). These cases illustrate how the two approaches differ in response style and hallucination tendency: DivPrune often yields broader and more descriptive captions but introduces hallucinated objects, whereas FasterVLM produces more conservative and focused descriptions

Moreover, Fig.14 presents a controlled experiment where the token budget is fixed at 64 and the ratio between DivPrune- and attention-selected tokens (DivPrune-to-Attention ratio, $R$) is varied in steps of 25%. We observe that the hallucinated objects frequently generated under pure DivPrune ($R = 0$) gradually diminish as the share of attention-selected tokens increases, disappearing entirely when $R \geq 50\%$. In parallel, the response style evolves from speculative and exploratory to more factual and reliable as $R$ increases, showing the stabilizing effect of attention-based token selection.

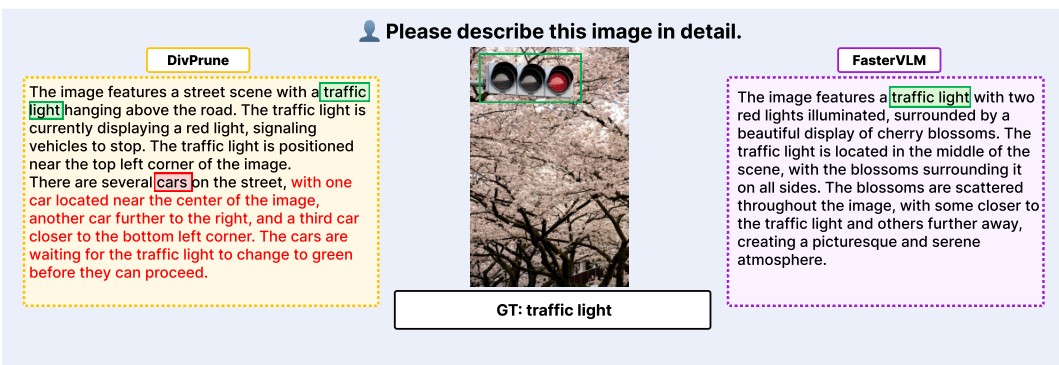

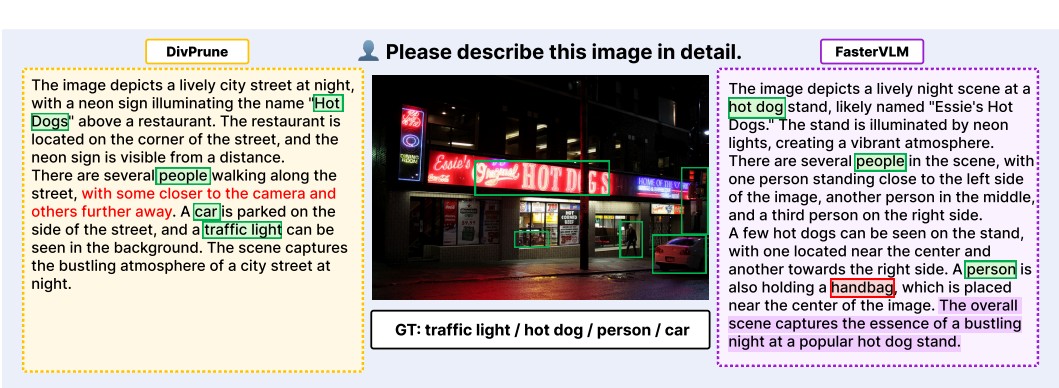

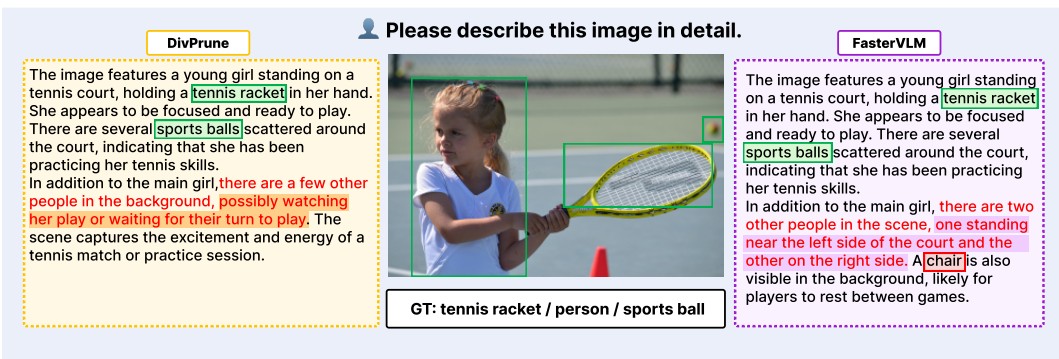

Figure 13: **CHAIR qualitative comparisons: FasterVLM vs. DivPrune (Set 2).** In the annotations, ■ GT Obj. and ■ Hallucinated Obj. label object words; ■ marks DivPrune's phrasing; ■ marks FasterVLM's phrasing; **red text** indicates incorrect phrases.

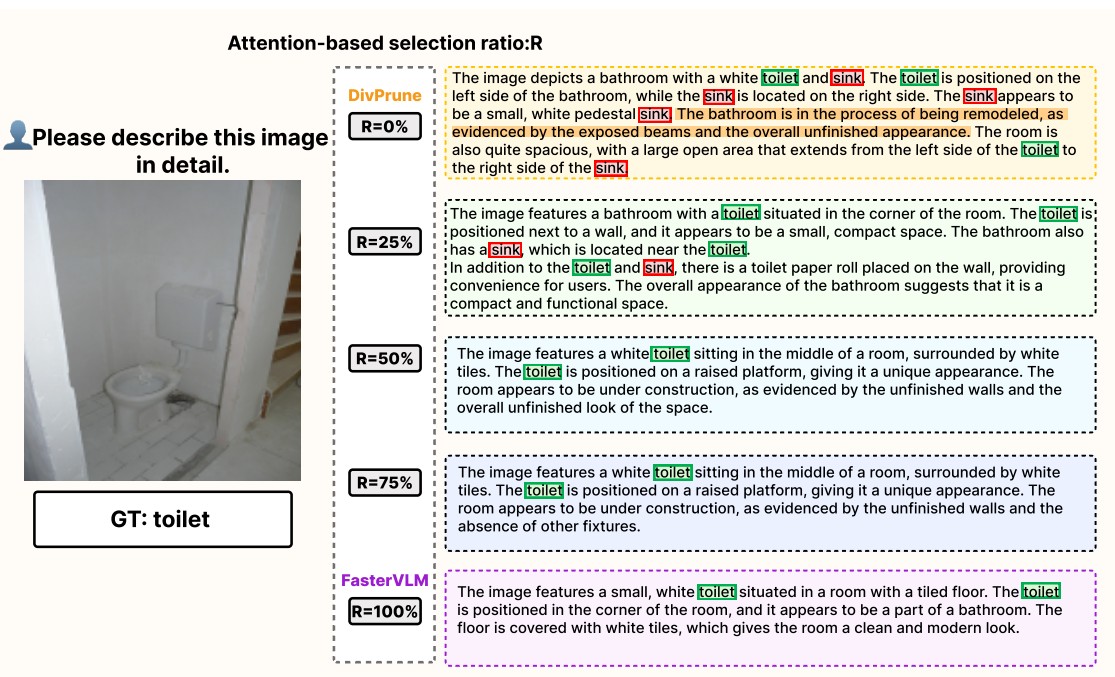

Figure 14: **Effect of varying the Attention-based selection ratio** $R$ **under a 64-token budget.** As $R$ increases, hallucinated objects produced by DivPrune are progressively suppressed, and the responses shift from exploratory to fact-oriented descriptions. In the annotations, ■GT obj. and ■Hallucinated obj. label object words; ▮ denotes DivPrune-specific phrasing.

## STATEMENT ON THE USE OF LARGE LANGUAGE MODELS

In the interest of transparency and in compliance with the ICLR 2026 guidelines, we report that a large language model (LLM) was used to assist in the refinement of this paper's text.

**Scope of Use.** The model's role was strictly limited to that of a writing assistant. Its contributions include:

- Correcting grammatical errors, spelling, and punctuation.
- Improving sentence structure and flow for enhanced clarity.
- Refining word choices for greater precision and conciseness.

