# OpenReview forum: "AgilePruner: An Empirical Study of Attention and Diversity for Adaptive Visual Token Pruning in Large Vision-Language Models"
_ICLR.cc/2026/Conference — ICLR 2026 Poster_

### Official Review · Reviewer_DUt4 · 2025-10-29

**Soundness:** 3
**Presentation:** 3
**Contribution:** 3
**Rating:** 4
**Confidence:** 4

**Summary:**

Large vision language models (LVLMs) generate hundreds of visual tokens that must be concatenated with textual tokens for processing, incurring quadratic time and memory costs. Many works have proposed attention‑based or diversity‑based pruning strategies to drop redundant visual tokens. Attention‑based methods keep tokens with high [CLS] attention scores but may select similar tokens, while diversity‑based methods maximize feature diversity but can discard important high‑attention tokens. The authors provide a systematic empirical study comparing these strategies across image types, using effective rank (erank) to quantify feature diversity and attention entropy to measure how concentrated the attention distribution is. They show that simple images (low erank / low entropy) benefit from attention‑based pruning, whereas complex images (high erank / high entropy) favor diversity‑based pruning. Furthermore, using the CHAIR hallucination benchmark, they find that attention‑based methods produce more conservative captions with lower hallucination scores, while diversity‑based methods capture more objects but hallucinate more. Motivated by these insights, the authors propose an adaptive token‑pruning framework. They explore tokens in descending order of attention scores and remove redundant tokens based on a similarity threshold that is adaptively set via a logarithmic mapping of erank. Simple images (low erank) use stricter thresholds; complex images use looser thresholds. Experiments show that the adaptive strategy maintains competitive performance.

**Strengths:**

S1. The paper systematically evaluates attention‑ and diversity‑based pruning across datasets, using effective rank and attention entropy to quantify image complexity. This analysis clarifies when each strategy excels.

S2. By measuring CHAIR metrics and recall, the authors show that diversity‑based pruning has higher recall but induces more hallucinations, while attention‑based pruning yields safer outputs.

S3. Experiments span nine multimodal benchmarks and the CHAIR hallucination dataset, showing the adaptive method maintains accuracy under aggressive pruning and reduces hallucination.

**Weaknesses:**

W1. The adaptive threshold function is heuristic and requires tuning coefficients alpha and beta. There is no principled justification or learning of this mapping.

W2. Computing effective rank involves singular‑value decomposition of token features, which may be expensive for large sequences. The paper does not quantify this overhead or compare it with more lightweight measures.

W3. Experiments are conducted exclusively on LLaVA‑1.5‑7B. Results may not generalize to other LVLMs or larger models, where token distributions or attention patterns differ.

W4. Improvements over baselines are relatively small (1–3% absolute), which may not justify the added complexity of computing erank and adaptive thresholds. Statistical significance is not reported.

W5. The paper notes that diversity‑based methods hallucinate more, but does not examine whether the adaptive method sometimes discards critical tokens (e.g., rare or small objects) or affects reasoning tasks beyond object captioning.

**Questions:**

Q1. How is erank computed in practice? Does it require an SVD for each image? What is the runtime overhead relative to token pruning, and how does it scale with image resolution and batch size?

Q2. Have you experimented with different mapping functions (e.g., linear or learned mappings) for tau_adaptive? How sensitive are the results to the chosen alpha and beta values across different datasets?

Q3. You use both erank and attention entropy to analyze image complexity, but only erank is used for threshold adaptation. Did you test using attention entropy or other metrics for adaptation?

Q4. For tasks requiring fine‑grained spatial reasoning or counting, does the adaptive pruning ever remove tokens critical to the answer? Can you provide qualitative examples where the method fails or underperforms?

---

> ### Author Response · Authors · 2025-11-21
> **Comment 1/5 for Reviewer DUt4**
>
> We thank the reviewer for the careful reading and thoughtful comments. Below, we address the reviewer’s questions and provide detailed explanations. The corresponding revisions in the manuscript are marked in blue for clarity. We hope that the responses below, along with the updated manuscript, address the reviewer’s concerns thoroughly. Due to character limits on OpenReview, our response has been divided into five parts for ease of readability. Thank you for your understanding.
>
> **Weakness 1, Question 2:** The adaptive threshold function is heuristic and requires tuning coefficients alpha and beta. There is no principled justification or learning of this mapping. Have you experimented with different mapping functions (e.g., linear or learned mappings) for tau_adaptive? How sensitive are the results to the chosen alpha and beta values across different datasets?
>
> **Response:**  We sincerely thank the reviewer for raising this important point. We fully agree that the initial logarithmic mapping, together with its coefficients (α, β), could appear heuristic in the absence of further justification. We appreciate the opportunity to clarify and strengthen this aspect in the revised manuscript.
>
> **1. Robustness of the mapping form.**
>
> Although our original design used a logarithmic function, subsequent analysis shows that the *specific mathematical form* is not what drives performance. we evaluated multiple alternative mappings—including linear, exponential, and step functions—and found that any **monotonically increasing** mapping with respect to erank yields nearly identical performance trends. This indicates that the core mechanism is not dependent on α or β, but rather on the **empirical principle that image complexity governs the preferred pruning paradigm** (simple images → attention-focused; complex images → diversity-focused).
>
> To assess whether these parameters substantially influence model behavior, we conducted a sensitivity study by varying the target threshold range. As summarized below, performance remained highly stable across settings, indicating that the adaptive mechanism does not rely on finely tuned hyperparameters:
>
> | Min Scaling | Max Scaling | MME (Perception) [1] | MME (Cognition) |
> | --- | --- | --- | --- |
> | 0.015 | 0.20 | 1413.3 | 325.3 |
> | 0.015 | 0.25 | 1413.6 | 315.3 |
> | 0.020 | 0.15 | 1418.6 | 326.0 |
> | 0.025 | 0.20 | 1416.4 | 320.7 |
>
> **2. Transition to a parameter-free, statistics-driven rule.**
>
> We fully agree that eliminating manually tuned coefficients improves both clarity and generalizability. For this reason, we refined our adaptive rule in the revised version to use a **statistics-driven formulation** based solely on data characteristics:
>
> $$
> \text{ratio} = \frac{\text{erank}\_{\text{input}}}{\text{erank}\_{\text{avg}}}
> $$
>
> - **Complex images (ratio > 1):** the threshold increases automatically, promoting more diverse token retention.
> - **Simple images (ratio < 1):** the threshold decreases, preserving focused evidence in high-attention regions.
>
> This refinement removes all manually tuned coefficients and yields a **fully parameter-free adaptive mechanism**. The rule adapts naturally to the distribution of erank in the dataset and directly reflects the empirical relationship uncovered in our analysis.
>
> We have updated **Section 4.3** to document this formulation clearly, and all main experimental results have been revised accordingly to reflect the updated formulation **in Sec 5.** We hope this resolves the reviewer’s concern by showing that the adaptive rule is not heuristic in nature, but a **principled, data-driven instantiation** of the underlying empirical finding.
>
> ---

---

> ### Author Response · Authors · 2025-11-21
> **Comment 2/5 for Reviewer DUt4**
>
> **Weakness 2,  Question 1:** Computing effective rank involves singular‑value decomposition of token features, which may be expensive for large sequences. The paper does not quantify this overhead or compare it with more lightweight measures. How is erank computed in practice? Does it require an SVD for each image? What is the runtime overhead relative to token pruning, and how does it scale with image resolution and batch size?
>
> **Response:**  We sincerely thank the reviewer for raising this important concern regarding the computational overhead of computing effective rank (erank). To address this point clearly, we conducted an additional analysis and compared the erank computation cost with the overall inference time.
>
> **1. Efficient computation via eigen decomposition.**
> Although a full SVD can be expensive, erank only requires the singular values of the token-feature matrix $X$. We use the well-known identity that the singular values of $X$ equal the square roots of the eigenvalues of $XX^\top$. This allows us to compute:
>
> $$
> C = XX^\top \in \\mathbb{R}^{N \\times N}, \\qquad
> S = \\sqrt{\\lambda(C)}, \\qquad
> p_i = \\frac{S_i}{\\sum_j S_j}, \\qquad
> \\text{erank}(X) = \\exp\\left( -\\sum_i p_i \\log p_i \\right),
> $$
>
> where $S$ denotes the singular values derived from the eigenvalues $\lambda(C)$, and $N$ is the number of visual tokens. Since LVLMs typically satisfy $N \ll D$ (e.g., 576 tokens vs. 4096 embedding dim), this formulation enables us to bypass the computationally expensive full SVD (complexity: $\mathcal{O}(ND^2)$). Instead, we rely on the eigendecomposition of the $N \\times N$ matrix, which reduces the complexity to $\mathcal{O}(N^2D + N^3)$, thereby achieving significantly higher efficiency.
>
> **2. Measured overhead in practice.**
>
> On LLaVA-1.5-7B (576 visual tokens, 4096-dim embeddings), the erank computation takes only ≈ 3.65 ms on a single RTX 4090 GPU at TextVQA dataset. [1] This corresponds to approximately 3.2% of the total inference time under the pruned configuration (115 ms per sample). Thus, the cost of computing erank is negligible in practical LVLM settings and does not hinder real-time usage.
>
> **3. Scaling with Image Resolution (Token Count)**
>
> If the image resolution—and thus the number of vision tokens $N$—increases, the covariance matrix becomes $N \times N$. Therefore, the theoretical worst-case complexity follows $\mathcal{O}(N^2D + N^3)$. However, in typical LVLM settings where $N \ll D$ (e.g., 576 tokens vs. 4096 dim), the cost is dominated by the $\mathcal{O}(N^2D)$ term. As a result, the effective cost grows approximately quadratically with the token count, keeping the overhead modest for standard resolutions.
>
> **4. Scaling with Batch Size**
>
> Since erank is computed independently for each image, the total computation increases approximately linearly with the batch size. We profiled this behavior on an RTX-4090, obtaining the following results:
>
> | **Batch Size** | 1 | 3 | 5 | 10 |
> | --- | --- | --- | --- | --- |
> | **erank overhead (ms)** | 3.65 | 10.21 | 16.84 | 33.40 |
>
> These measurements confirm the expected near-linear scaling trend.
>
> **5. Attention entropy as a Lightweight Substitute for erank**
>
> We also explored whether attention entropy—another complexity indicator—could replace erank in the adaptive rule. The two metrics exhibit a moderate positive correlation (Pearson ρ = 0.63 in full MME Dataset), indicating that entropy captures similar complexity trends.
>
> Importantly, entropy is extremely inexpensive to compute. On the same hardware used for our main experiments:
>
> - entropy computation runtime ≈ 0.10 ms
> - total inference time ≈ 115 ms
> - overhead ≈ 0.08%, which is effectively negligible.
>
> We applied the same adaptive rule to both metrics and compared their performance:
>
> | Retain tokens | Metric | GQA | ScienceQA | POPE | MME | TextVQA |
> | --- | --- | --- | --- | --- | --- | --- |
> | **128** | Erank-based | 59.4 | 68.6 | 87.4 | 1748 | 57.0 |
> | **128** | Entropy-based | 59.4 | 68.6 | 87.0 | 1721 | 56.9 |
> | **64** | Erank-based | 57.4 | 68.6 | 84.1 | 1703 | 56.0 |
> | **64** | Entropy-based | 57.7 | 68.7 | 83.2 | 1690 | 56.0 |
>
> As shown, entropy-based adaptation exhibits consistent behavior and yields moderately similar accuracy across benchmarks, confirming that it is a viable lightweight proxy when erank computation must be minimized.
>
> **6. Comparison with prior lightweight alternatives.**
>
> While some recent in-LLM pruning methods (e.g., PDrop, SparseVLM) avoid erank computation, they perform pruning *inside* the decoder. As a result, early decoder layers still process all 576 tokens, and their end-to-end latency is higher despite similar FLOPs. In contrast, our pre-pruning approach reduces the sequence length before entering the LLM, enabling consistent compute and memory savings across all layers.
>
> We have included the full runtime breakdown, memory footprint analysis, and details of our implementation in **Appendix A**.
>
> ---

---

> ### Author Response · Authors · 2025-11-21
> **Comment 3/5 for Reviewer DUt4**
>
> **Weakness 3:** Experiments are conducted exclusively on LLaVA‑1.5‑7B. Results may not generalize to other LVLMs or larger models, where token distributions or attention patterns differ.
>
> **Response:**  We thank the reviewer for raising this important point. We fully agree that token distributions and attention patterns can vary substantially across LVLM architectures, and we appreciate the opportunity to demonstrate that our findings generalize beyond LLaVA-1.5-7B.
>
> **1. Validation on larger LLaVA models.**
>
> To ensure scalability within the same model family, we extended our experiments to **LLaVA-1.5-13B [1]** and **LLaVA-NeXT-7B. [2]**. These results are included in Appendix B.1. Across both larger and more recent LLaVA variants, our adaptive strategy continues to show the same image-complexity–dependent behavior and improves performance over fixed-strategy pruning.
>
> **2. Generalization across architectures (Qwen-VL).**
>
> To directly address the reviewer’s concern about differing attention patterns, we additionally evaluated our method on **Qwen2.5-VL-7B [3]**, which has a substantially different architecture and token distribution compared to LLaVA. The adaptive rule yields consistent gains on Qwen-VL as well, suggesting that our empirical findings capture model-agnostic behavior rather than LLaVA-specific patterns. The extended results on Qwen-VL have been newly included in **Appendix B.1.**
>
> Representative results:
>
> | Method | Retain Tokens | TextVQA | ChartQA  | AI2D  | OCRBench  | MME (Perc.)  | MME (Cogn.) | MMB-EN | MMB-CN | Rel. |
> | --- | --- | --- | --- | --- | --- | --- | --- | --- | --- | --- |
> | Qwen2.5-VL-7B | 1225 | 82.1 | 77.5 | 83.0 | 84.1 | 1691 | 642 | 83.2 | 83.2 | 100.0% |
> | DivPrune (CVPR'25) | 512 | 79.7 | 72.0 | 81.9 | 78.5 | 1704 | 620 | 81.8 | 81.0 | 96.8% |
> | **Ours** | 512 | 79.8 | 71.5 | 82.1 | 78.5 | 1700 | 630 | 81.3 | 80.8 | **96.9%** |
> | DivPrune (CVPR'25) | 256 | 75.2 | 62.9 | 79.7 | 66.4 | 1703 | 542 | 80.3 | 79.8 | 90.3% |
> | **Ours** | 256 | 74.2 | 62.4 | 80.5 | 65.8 | 1670 | 630 | 80.9 | 80.3 | **92.1%** |
>
> **3. Adaptation to varying token distributions.**
>
> Our refined adaptive rule does not rely on absolute similarity thresholds. Instead, it computes the pruning threshold using the **ratio between the input image’s erank and the dataset’s average erank**. This formulation allows the rule to automatically *self-calibrate* to the characteristics of each model—whether the token embeddings are compact (e.g., LLaVA-7B), more dispersed (e.g., Qwen-VL), or scale-dependent (e.g., 13B variants). The consistent improvements observed across these models indicate that the adaptive mechanism generalizes reliably, even when token distributions and attention signatures differ.
>
> We have added these results and clarifications to the revised appendix. We sincerely appreciate the reviewer’s comment, which helped us strengthen the generalization claims of our work.
>
> Reference:
>
> [1] Liu, Haotian, et al. "Improved baselines with visual instruction tuning." *Proceedings of the IEEE/CVF conference on computer vision and pattern recognition*. 2024.
>
> [2] Li, Bo; Zhang, Kaichen; Zhang, Hao; Guo, Dong; Zhang, Renrui; Li, Feng; Zhang, Yuanhan; Liu, Ziwei; Li, Chunyuan. *“LLaVA-NeXT: Stronger LLMs Supercharge Multimodal Capabilities in the Wild.”* 10 May 2024. URL: [https://llava-vl.github.io/blog/2024-05-10-llava-next-stronger-llms/](https://llava-vl.github.io/blog/2024-05-10-llava-next-stronger-llms/?utm_source=chatgpt.com)
>
> [3] Bai, Shuai, et al. "Qwen2. 5-vl technical report." *arXiv preprint arXiv:2502.13923* (2025).
>
> ---

---

> ### Author Response · Authors · 2025-11-21
> **Comment 4/5 for Reviewer DUt4**
>
> **Weakness 4:** Improvements over baselines are relatively small (1–3% absolute), which may not justify the added complexity of computing erank and adaptive thresholds. Statistical significance is not reported.
>
> **Response:**  We sincerely thank the reviewer for this thoughtful observation. We address the two concerns—(1) whether the observed improvements justify the computational overhead, and (2) whether the gains are statistically meaningful—below.
>
> **1. Computational overhead of erank and adaptive thresholding.**
>
> To quantify the cost more precisely, we evaluated FLOPs, latency, and memory under identical conditions using an RTX 4090 GPU. The table below summarizes the comparison with representative pre-pruning and in-LLM pruning baselines:
>
> | Method | Retain | FLOPs (T) | Latency (ms/sample) | GPU Mem (GB) | Acc (%) |
> | --- | --- | --- | --- | --- | --- |
> | Vanilla (LLaVA-1.5-7B) | 576 | 3.14 | 172 | 13.60 | 58.2 |
> | PDrop (CVPR’25) | 64 | 0.51 | 128 | 13.30 | 55.0 |
> | SparseVLM (ICML’25) | 64 | 0.52 | 129 | 16.26 | 55.2 |
> | DivPrune (CVPR’25) | 64 | 0.48 | 110 | 13.30 | 55.8 |
> | VisPruner (ICCV’25) | 64 | 0.48 | 115 | 13.30 | 55.4 |
> | **Ours (Pre-Pruning)** | 64 | 0.48 | 115 | 13.30 | 56.0 |
>
> Computing erank takes **≈ 3.65 ms** per sample, corresponding to only **~3.2%** of the pruned model’s total inference time (115 ms). Furthermore, because our method performs pruning **before** the LLM, all subsequent decoder layers immediately benefit from the shorter sequence. This contrasts with in-LLM pruning such as PDrop or SparseVLM, where early decoder layers still process all 576 tokens, leading to higher latency despite similar FLOPs. Memory usage also remains identical to standard pre-pruning baselines.
> These results indicate that the proposed adaptivity does **not** introduce meaningful runtime or memory overhead in practice.
>
> **2. Statistical significance of the reported performance gains.**
>
> All evaluations were conducted with **temperature = 0**, following widespread practice in LVLM pruning research. Under this deterministic setting, repeated runs yield identical outputs, and variance due to sampling noise is eliminated. As a result, the observed 1–3% absolute gains are **stable and repeatable**, not the result of stochastic fluctuations.
>
> We also highlight that in the context of token pruning—where the goal is to maintain performance under strict FLOPs/token constraints—**preserving or modestly improving accuracy while reducing computation is a meaningful improvement**. Among all pre-pruning baselines, our method consistently exhibits the **smallest accuracy drop** relative to the full-token model under identical budgets.
>
> Overall, within the same computational and memory footprint as prior methods, our adaptive strategy provides **consistent, reproducible, and practically significant accuracy gains** across diverse benchmarks. We have expanded the efficiency analysis and clarified the deterministic evaluation protocol in the revised appendix. We sincerely appreciate the reviewer’s comment, which allowed us to present these results more clearly.
>
> ---
>
> **Question 3:** You use both erank and attention entropy to analyze image complexity, but only erank is used for threshold adaptation. Did you test using attention entropy or other metrics for adaptation?
>
> **Response:**
>
> We thank the reviewer for the insightful question regarding whether attention entropy or other complexity metrics could also be used for adaptive thresholding.
>
> To directly address this, we conducted an additional experiment in which attention entropy was used in place of erank to compute the adaptive pruning ratio, following the exact same strategy as our erank-based rule (**Eq. 6 in Section 4.3**). the reference value entropy average was computed once using the LLaVA training set.
>
> The results show that entropy-based adaptation behaves very similarly to erank-based adaptation across benchmarks.
>
> | Retain tokens | Method | GQA | ScienceQA | POPE | MME | TextVQA | Rel. |
> | --- | --- | --- | --- | --- | --- | --- | --- |
> | 576 | LLaVA-1.5-7B | 61.9 | 69.5 | 85.9 | 1862 | 58.2 | 100.0% |
> | **128** | Entropy-based | 59.4 | 68.6 | 87.0 | 1721 | 56.9 | 98.13% |
> | **128** | erank-based | 59.4 | 68.6 | 87.4 | 1748 | 57.0 | 98.06% |
> | **64** | Entropy-based | 57.7 | 68.7 | 83.2 | 1690 | 56.0 | 95.97% |
> | **64** | erank-based | 57.4 | 68.6 | 84.1 | 1703 | 56.0 | 95.84% |
>
> Overall, entropy-based adaptation yields nearly identical trends to erank-based adaptation, with small variations but no systematic advantage.
>
> We believe this similarity arises because attention entropy and erank are moderately correlated in our analysis (Pearson correlation 0.63 measured over the full MME dataset). Both metrics reflect the dispersion of visual information, though erank additionally captures feature-space geometry via the singular-value spectrum.
>
> We report the full correlation study and additional entropy-based adaptation results in **Appendix B.2**.
>
> ---

---

> ### Author Response · Authors · 2025-11-21
> **Comment 5/5 for Reviewer DUt4**
>
> **Question 4:** For tasks requiring fine‑grained spatial reasoning or counting, does the adaptive pruning ever remove tokens critical to the answer? Can you provide qualitative examples where the method fails or underperforms?
>
> **Response:**
>
> Thank you for raising this important question. As the reviewer pointed out, adaptive pruning may potentially remove critical tokens in reasoning tasks—such as counting or fine-grained spatial reasoning—that rely on specific visual details. We address this concern below.
>
> First, our work evaluates a total of nine multimodal benchmarks (MME, MMBench-En/Cn, SQA, GQA, VQAv2, POPE, VizWiz, TextVQA), all of which require complex reasoning, knowledge, and precise visual understanding beyond simple object captioning. Across these diverse benchmarks, our adaptive method consistently achieves equal or superior performance compared to existing pruning approaches, providing quantitative evidence that the adaptive strategy does not excessively remove important tokens necessary for reasoning.
>
> In addition, following the reviewer’s suggestion, we not only analyzed hallucinations in image captioning but also conducted qualitative comparisons between diversity-based and attention-based methods on more fine-grained reasoning tasks such as counting and spatial relation inference. Furthermore, we qualitatively evaluated the adaptive method by comparing how all three pruning strategies select tokens and generate open-ended responses. These qualitative results are presented in **Appendix F.1** and **Figure 9**.
> Examples 1 and 4 show that attention-based pruning, due to its narrow focus on specific regions, may overlook certain objects and produce incorrect answers. In contrast, diversity-based pruning preserves broader spatial information, captures more objects, and tends to provide more detailed explanations, often resulting in more accurate predictions in such cases.
> However, in Examples 2 and 5, the opposite occurs: attention-based pruning correctly identifies the number of objects by focusing on the main instances, whereas diversity-based pruning—because of its dispersed selection—incorrectly predicts the presence of additional objects, causing hallucination.
> The adaptive method exhibits intermediate characteristics between attention and diversity. In Examples 1 and 4, it focuses on a narrower area than the diversity-based method and fails at counting; however, all three methods struggle in these particular challenging cases. In contrast, in Examples 2 and 5, the adaptive method accurately predicts the correct count alongside the attention-based approach, while the diversity-based method hallucinates additional objects.
>
> A similar pattern appears in spatial reasoning.
> As illustrated in Examples 3 and 6, the diversity-based method uses a wide range of spatial cues and generates rich relational descriptions but may infer relations that do not actually exist. Conversely, the attention-based method provides stable, object-centered location reasoning but with less comprehensive relational coverage. The adaptive method again lies between the two, producing descriptions that include the main objects while reducing the hallucinations observed in the diversity-based approach, leading to more stable spatial reasoning.
>
> Finally, to directly examine whether the adaptive method misses small or rare objects, we compared the three pruning strategies on an Existence task that asks whether a specific object is present in the image (**Appendix F.1, Fig. 10**).
> As seen in Example 1, the adaptive method successfully preserved the necessary tokens to identify a small object that the attention-based method overlooked.
> Similarly, in Example 2, where both attention-based and diversity-based methods failed to detect a very small object, the adaptive method retained the relevant information and produced the correct answer. These results demonstrate that the adaptive strategy reliably preserves important tokens and remains effective even when small objects serve as critical cues for reasoning.
>
> Taken together, across diverse reasoning tasks, the adaptive method demonstrates stable and reliable performance—not only in quantitative benchmarks but also through qualitative examples.
>
> ---

---

> ### Comment · Reviewer_DUt4 · 2025-11-22
>
> Thank you for the thorough rebuttal and additional experiments. While several concerns remain, the clarifications meaningfully reduce my earlier concerns. I increased my score.
>
> My remaining reservations are:
> - The adaptive rule is intentionally minimal, and the primary novelty still lies in empirical characterization rather than a substantial methodological advance.
> - Although consistent, the improvements over existing pruning baselines remain modest, and do not by themselves constitute a strong algorithmic step forward.
> - The qualitative examples help, but a systematic understanding of when the adaptive method fails—especially on fine-grained reasoning—remains incomplete.

---

> ### Author Response · Authors · 2025-11-22
>
> Thank you very much for your thoughtful follow-up. We appreciate the important observation that, in the initial submission, the distinction between the *empirical contribution* and the *methodological instantiation* was not sufficiently explicit. During the rebuttal period, we revised the **main empirical analysis (Sec. 4)** and relevant appendices to address concerns raised by all reviewers. Following your specific follow-up comment, we further refined the **abstract** and **introduction** (highlighted in blue in the updated PDF) to make our intended framing fully explicit.
>
> More specifically, the revised manuscript now foregrounds three empirical findings that, to our knowledge, have not been documented in prior LVLM pruning literature:
> (1) a quantitative erank-based analysis showing that many “diversity-aware’’ pruning methods preserve less feature diversity than intended,
> (2) a strong and previously unreported link between retained-token diversity and **hallucination behavior**, and
> (3) evidence that **image complexity** consistently predicts whether attention-based or diversity-based pruning is preferable, resolving several inconsistencies observed across datasets and pruning paradigms.
>
> To more transparently reflect these contributions, we **clarified** in both the abstract and introduction that the primary novelty of our work lies in uncovering these empirical principles. The adaptive rule is intentionally minimal and presented only as a concise instantiation of the empirical insight—not as a standalone methodological novelty. Importantly, the fact that such adaptive mechanism consistently improves multiple pruning paradigms (BAT, VisPruner, DivPrune+FasterVLM) provides strong evidence that the underlying empirical principle is robust, fundamental, and model-agnostic.
>
> This framing aligns with the evolution of VLM token-pruning research: influential works such as FastV, FasterVLM, VisionZip, and DivPrune begin from an empirical observation—e.g., attention sparsity or diversity collapse—and validate it through deliberately lightweight mechanisms. Our work follows this paradigm but extends it by identifying a broader LVLM-specific principle that unifies and explains these behaviors.
>
> Regarding your request for a systematic characterization of failure cases in fine-grained reasoning, we have added a structured failure-mode summary in **Appendix F.2**. This analysis identifies two primary conditions under which the adaptive method may underperform:
>
> 1. **Low-erank images with many small or widely distributed objects**:
>    In such cases, the adaptive method behaves similarly to attention-based pruning—focusing on localized regions.
>    It succeeds when objects cluster in one area (Fig. 11a), but fails when many objects are scattered across the scene (Fig. 11b).
>
> 2. **High-erank images where crucial evidence is locally concentrated**:
>    Here the method behaves more like diversity-based pruning, selecting spatially dispersed tokens.
>    It succeeds when objects are broadly distributed (Fig. 11c), but fails when key cues lie in a highly localized region (Fig. 11d), as the wide-spread selection dilutes essential evidence.
>
> Together, these cases clarify when and why the adaptive rule struggles in fine-grained reasoning tasks, and we hope the expanded characterization fully addresses your concern.
>
> Thank you again for your constructive feedback. Your follow-up comment directly strengthened the clarity, framing, and contribution of the paper.

---

### Official Review · Reviewer_g8H8 · 2025-10-30

**Soundness:** 3
**Presentation:** 4
**Contribution:** 2
**Rating:** 8
**Confidence:** 4

**Summary:**

This paper analyzes the limitations of existing token pruning methods designed to improve the efficiency of LVLMs and proposes a novel token pruning framework to address them. The authors employ attention score entropy and effective rank to examine the characteristics of each image token, and their proposed method is designed to adaptively apply pruning based on the specific properties of the image.

**Strengths:**

1. The properties and limitations of attention-based and diversity-based token pruning methods have not been clearly studied before. This paper provides both quantitative and qualitative analyses of these issues, making the argument convincing.
2. Based on the observed trends, the authors propose an intuitive token pruning method.
3. The proposed approach demonstrates high reliability by balancing the performance–stability trade-off between attention-based and diversity-based methods, which is further supported by extensive experimental results on various benchmarks.

**Weaknesses:**

1. In Table 1, the difference in entropy scores seems small. Additional explanation regarding the scale of these scores would be helpful.
2. The performance seems sensitive to the similarity threshold. Although the authors attempt to make the design adaptive using the effective rank (erank), parameter tuning (alpha, beta) may still limit the generalizability of the proposed method.

**Questions:**

1. The values in Table 1 differ from the descriptions in Section 4.1 (OCR: 4.39, 49, POPE: 4.90, 109).
2. After token pruning in the Table 1 experiments, how does the attention entropy change?

---

> ### Author Response · Authors · 2025-11-21
> **Comment 1/2 for Reviewer g8H8**
>
> We thank the reviewer for the careful reading and thoughtful comments. Below, we address the reviewer’s questions and provide detailed explanations. The corresponding revisions in the manuscript are marked in blue for clarity. We hope that the responses below, along with the updated manuscript, address the reviewer’s concerns thoroughly. Due to character limits on OpenReview, our response has been divided into two parts for ease of readability. Thank you for your understanding.
>
> **Weakness 1:**  In Table 1, the difference in entropy scores seems small. Additional explanation regarding the scale of these scores would be helpful.
>
> **Response:**  We thank the reviewer for raising this helpful point. We agree that the entropy differences reported in **Table 4 (Table 1 in the original submission)** may appear small, and we have clarified the scale of this metric more explicitly in the revised manuscript.
>
> To provide better context, we examined the typical entropy range of the attention distributions produced by the CLIP-L vision encoder [1] (576 visual tokens) on the MME dataset [2]. As shown in the histograms added to the **appendix D (Figure 7)**, attention entropy naturally lies within a **very narrow interval of approximately 4.0–5.4** across diverse images. For example:
>
> - **Top 25%  entropy:** ~5.00
> - **mean entropy:** ~4.80
> - **Bottom 25% entropy:** ~4.67
>
> Because the entropy distribution itself occupies such a limited range, even differences of **0.3–0.5** correspond to meaningful changes in attention concentration. In contrast, **erank spans a much wider numerical scale**, which can make the two metrics appear inconsistent in magnitude when viewed side-by-side.
>
> To make this clearer for readers, we have added descriptive statistics and histogram visualizations (Figure 7) in **Appendix D**, along with a brief explanation in the main text describing how to interpret the respective scales of entropy and erank. We hope this improves the readability of Table 4 and helps contextualize the significance of the reported entropy differences.
>
> References:
>
> [1] Radford, Alec, et al. "Learning transferable visual models from natural language supervision." *International conference on machine learning*. PmLR, 2021.
>
> [2] Fu, Chaoyou, et al. "Mme: A comprehensive evaluation benchmark for multimodal large language models." *The Thirty-ninth Annual Conference on Neural Information Processing Systems Datasets and Benchmarks Track*. 2025.
>
> ---
>
> **Weakness 2:** The performance seems sensitive to the similarity threshold. Although the authors attempt to make the design adaptive using the effective rank (erank), parameter tuning (alpha, beta) may still limit the generalizability of the proposed method.
>
> **Response:**  We appreciate the reviewer’s thoughtful comment regarding the sensitivity of the similarity threshold and the potential impact of the parameters used in our adaptive design. We agree that ensuring generalizability is important, and we have clarified this aspect more explicitly in the revised manuscript.
>
> **First,** our extended sensitivity study shows that the performance remains **highly stable** even when the scaling parameters vary across a reasonably wide range. The fluctuations in accuracy are small, indicating that the adaptive mechanism is not dependent on precise parameter values.
>
> | Min Scaling | Max Scaling | MME (Perception) [1] | MME (Cognition) |
> | --- | --- | --- | --- |
> | 0.015 | 0.20 | 1413.3 | 325.3 |
> | 0.015 | 0.25 | 1413.6 | 315.3 |
> | 0.020 | 0.15 | 1418.6 | 326.0 |
> | 0.025 | 0.20 | 1416.4 | 320.7 |
>
> **Second,** To further reduce dependence on explicit coefficients and make the mechanism more transparent, we refined the adaptive thresholding rule in the revised manuscript. The new formulation uses a **dataset-driven ratio**:
>
> $$
> \text{ratio} = \frac{\text{erank}\_{\text{input}}}{\text{erank}\_{\text{avg}}}
> $$
>
> - **Complex images (ratio > 1):** threshold increases, enabling more aggressive pruning to ensure diversity.
> - **Simple images (ratio < 1):** threshold decreases, preserving focused and discriminative tokens.
>
> This strategy removes the need for manually tuned parameters and directly operationalizes the empirical finding that **image complexity predicts the preferred pruning paradigm**.
>
> We also verified that this simplified rule behaves consistently across multiple LVLM architectures (LLaVA-1.5-7B/13B, LLaVA-NeXT-7B, Qwen-VL), suggesting that the adaptive mechanism is **robust and architecture-agnostic**. The results for these additional models are provided in **Appendix B.1**.
>
> We have updated **Section 4.3** to document this formulation clearly, and all main experimental results have been revised accordingly to reflect the updated formulation **in Sec 5.** and we hope this clarification alleviates the reviewer’s concern about parameter sensitivity.
>
> ---

---

> ### Author Response · Authors · 2025-11-21
> **Comment 2/2 for Reviewer g8H8**
>
> **Question 1:** The values in Table 1 differ from the descriptions in Section 4.1 (OCR: 4.39, 49, POPE: 4.90, 109).
>
> **Response:**  We sincerely thank the reviewer for carefully checking the consistency between **Section 4.2 (Section 4.1 in the original submission)** and **Table 4 (Table 1 in the original submission)**. Upon reviewing the numbers, we found that the values mentioned in the text for **OCR [1]** and **POPE [2]** were mistakenly copied from the **Text Translation** and **Position** subsets, respectively, rather than from the correct OCR/POPE results.
>
> The entries shown in Table 4 for OCR and POPE are the correct ones and match the underlying experimental outputs. We have updated the corresponding descriptions in Section 4.2 so that all numerical values in the main text are now fully consistent with Table 4.
>
> We appreciate the reviewer’s attention to detail, which helped us correct this oversight in the revised manuscript.
>
> References:
>
> [1] Fu, Chaoyou, et al. "Mme: A comprehensive evaluation benchmark for multimodal large language models." *The Thirty-ninth Annual Conference on Neural Information Processing Systems Datasets and Benchmarks Track*. 2025.
>
> [2] Li, Yifan, et al. "Evaluating object hallucination in large vision-language models." *arXiv preprint arXiv:2305.10355* (2023).
>
> ---
>
> **Question 2:** 	After token pruning in the Table 1 experiments, how does the attention entropy change?
>
> **Response:**
> Thank you for the question. In addition to attention entropy, we also evaluated effective rank (erank) to more comprehensively analyze the behavior of the two pruning strategies. After pruning, we observe a consistent difference in how attention entropy changes. As shown in the table below, the attention-based strategy yields higher post-pruning entropy across all datasets (MME, ScienceQA, and POPE). This arises because attention-based pruning selects the top-k tokens and removes the long tail of low-attention tokens. Consequently, the retained tokens have uniformly higher attention scores, which produces a flatter normalized distribution and therefore higher entropy
>
> Meanwhile, erank directly reflects how much diversity is preserved after pruning. Across all datasets, the diversity-based method consistently shows higher erank than the attention-based approach, because it selects a more varied set of tokens. This indicates that diversity-oriented pruning preserves a wider range of visual information.
>
> | Metric | OCR  | Numerical Cal.  | Text Translation | ScienceQA  | Position | Scene | Count | POPE |
> | --- | --- | --- | --- | --- | --- | --- | --- | --- |
> | **Att. entropy** | 4.61 | 4.47 | 4.39 | 4.45 | 4.90 | 4.86 | 4.82 | 4.87 |
> | **after entropy (att. based)** | 3.55 | 3.54 | 3.31 | 3.45 | 3.46 | 3.42 | 3.48 | 3.48 |
> | **after entropy (div. based)** | 2.69 | 2.87 | 2.65 | 2.73 | 2.82 | 2.73 | 2.77 | 2.81 |
> | **erank** | 78 | 59 | 49 | 74 | 109 | 103 | 102 | 106 |
> | **after erank (att. based)** | 11.4 | 8.46 | 6.89 | 10.19 | 15.38 | 12.46 | 13.13 | 13.03 |
> | **after erank (div. based)** | 15.6 | 11.5 | 10.1 | 15.68 | 17.60 | 16.50 | 17.30 | 17.31 |
>
> ---

---

### Official Review · Reviewer_2wEG · 2025-11-03

**Soundness:** 2
**Presentation:** 2
**Contribution:** 1
**Rating:** 2
**Confidence:** 4

**Summary:**

This paper presents an empirical study on visual token pruning in large vision-language models (LVLMs). The authors analyze two main pruning paradigms: attention-based and diversity-based, and characterize their behavior using attention entropy and effective rank (erank) metrics, respectively. They show that attention-based pruning works better on simple images with concentrated features, while diversity-based pruning performs better on complex images with distributed features. They further study hallucination behavior and propose an adaptive pruning framework that adjusts token similarity thresholds according to image complexity, claiming improvements over several baselines across multiple benchmarks.

**Strengths:**

1.	The paper provides a thorough and systematic empirical comparison between attention- and diversity-based pruning strategies, which is less explored in depth before.
2.	The adoption of effective rank and attention entropy as quantitative measures for image complexity is conceptually reasonable.

**Weaknesses:**

1.	The proposed adaptive thresholding strategy is relatively simple and heuristic (a logarithmic mapping between erank and threshold). It does not provide strong methodological or theoretical innovation beyond straightforward empirical observations.
2.	The proposed adaptive thresholding strategy introduces several hyperparameters, notably the scaling coefficients and other implementation choices. These parameters may influence pruning behavior, yet the paper does not provide a clear justification or sensitivity analysis for them.
3.	Prior token pruning works (e.g., DivPrune, VisionZip, PruMerge+) already explore balancing attention and diversity. This paper’s “adaptive” version appears incremental and lacks a compelling distinction.

**Questions:**

See weaknesses.

---

> ### Author Response · Authors · 2025-11-21
> **Comment 1/3 for Reviewer 2wEG**
>
> We thank the reviewer for the careful reading and thoughtful comments. Below, we address the reviewer’s questions and provide detailed explanations. The corresponding revisions in the manuscript are marked in blue for clarity. We hope that the responses below, along with the updated manuscript, address the reviewer’s concerns thoroughly. Due to character limits on OpenReview, our response has been divided into three parts for ease of readability. Thank you for your understanding.
>
> **Weakness 1:** The proposed adaptive thresholding strategy is relatively simple and heuristic (a logarithmic mapping between erank and threshold). It does not provide strong methodological or theoretical innovation beyond straightforward empirical observations.
>
> **Response:**   We sincerely thank the reviewer for raising this point. We agree that the adaptive rule we use is intentionally simple. However, the goal of our work is not to propose a complex new pruning mechanism, but rather to provide a **principled and empirically grounded understanding** of why attention-based and diversity-based pruning behave differently in LVLMs, and to show how this understanding can be translated into a practical rule.
>
> As clarified in the revised Section 4, our analysis reveals several **non-obvious and previously unreported empirical patterns** linking:
>
> - the diversity of the retained token sets (measured by erank),
> - hallucination behavior, and
> - image-level complexity.
>
> These observations suggest that attention-focused and diversity-focused pruning succeed under **different structural conditions of the input**, a phenomenon that, to our knowledge, has not been examined in prior LVLM pruning works—including hybrid approaches such as PruMerge+ [1], VisionZip [2], and VisPruner [3]. In this sense, our adaptive rule is not positioned as methodological innovation on its own, but as a **minimal instantiation** of the empirical principle uncovered by our analysis.
>
> In response to the reviewer’s comment, we also carried out **additional experiments during the rebuttal period** to verify that this empirical principle is not limited to our own pruning strategy. Specifically,  Applying the simple erank-guided linear mapping to existing methods—which originally rely on fixed importance–diversity ratios, such as BAT [4] and VisPruner [5]—consistently improves their performance, while applying the inverse adjustment led to performance drops.
>
> A summary of these results is provided below:
>
> | Method | Retain tokens | GQA | ScienceQA | POPE | MME | Rel. |
> | --- | --- | --- | --- | --- | --- | --- |
> | **LLaVA-1.5-7B** | 576 | 61.9 | 69.5 | 85.9 | 1862 | 100.0% |
> | BAT (CVPR’23) | 128 | 58.6 | 69.3 | 85.3 | 1737 | 96.75% |
> | BAT + Inverse adaptation | 128 | 58.4 (-0.2) | 69.1 (-0.2) | 85.0 (-0.3) | 1734 (-3) | 96.46% (-0.19) |
> | **BAT + Ours** | 128 | **58.8 (+0.2)** | **69.4 (+0.1)** | **86.8 (+0.9)** | **1782 (+45)** | **97.91% (+1.11)** |
> | Vispruner (ICCV’25) | 128 | 58.2 | 69.0 | 84.6 | 1768 | 96.72% |
> | Vispruner + Inverse adaptation | 128 | 57.9 (-0.3) | 68.8 (-0.2) | 85.3 (+0.7) | 1744 (-24) | 96.35% (-0.37) |
> | **Vispruner + Ours** | 128 | **58.6 (+0.4)** | **69.1 (+0.1)** | **85.5 (+0.9)** | **1787 (+19)** | **97.32% (+0.60)** |
> | BAT (CVPR’23) | 64 | 56.6 | 68.8 | 81.5 | 1683 | 93.91% |
> | BAT + Inverse adaptation | 64 | 55.8 (-0.8) | 68.7 (-0.1) | 79.7 (-1.8) | 1602 (-81) | 91.98% (-1.93) |
> | **BAT + Ours** | 64 | **56.9 (+0.3)** | **69.3 (+0.5)** | **83.5 (+2.0)** | **1692 (+9)** | **94.85% (+0.94)** |
> | Vispruner (ICCV’25) | 64 | 55.4 | 69.1 | 80.4 | 1650 | 92.78% |
> | Vispruner + Inverse adaptation | 64 | 55.2 (-0.2) | 68.9 (-0.2) | 80.2 (-0.2) | 1643 (-7) | 91.83% (-0.95) |
> | **Vispruner + Ours** | 64 | **55.9 (+0.5)** | **69.3 (+0.2)** | **81.5 (+1.1)** | **1671 (+21)** | **93.76% (+0.98)** |
>
> We also observe the same trend for heterogeneous mixtures (DivPrune + FasterVLM). this combination also benefited from the erank-guided rule, confirming that the discovered principle is method-agnostic and captures general LVLM pruning behavior rather than a heuristic tied to a single model.
>
> | Method | Retain tokens | GQA | ScienceQA | POPE | MME | Rel. |
> | --- | --- | --- | --- | --- | --- | --- |
> | **LLaVA-1.5-7B** | 576 | 61.9 | 69.5 | 85.9 | 1862 | 100.0% |
> | Divprune (CVPR’25)   | 128 | 59.4 | 68.6 | 87.0 | 1707 | 96.90% |
> | FasterVLM (ArXiv’24) | 128 | 57.85 | 68.47 | 83.2 | 1757 | 95.80% |
> | Divprune+FasterVLM (fixed)  | 128 | 58.35 | 68.52 | 84.9 | 1768 | 96.66% |
> | **Divprune+FasterVLM (adaptive)** | **128** | **58.87** | **68.82** | **85.99** | **1787** | **97.55%** |
> | Divprune (CVPR’25)   | 64 | 57.5 | 68.0 | 85.5 | 1615 | 94.25% |
> | FasterVLM   (ArXiv’24) | 64 | 55.03 | 69.0 | 77.4 | 1665 | 91.91% |
> | Divprune+FasterVLM (fixed) | 64 | 56.75 | 68.57 | 82.2 | 1681 | 94.08% |
> | **Divprune+FasterVLM (adaptive)** | **64** | **57.23** | **68.86** | **83.5** | **1690** | **94.87%** |

---

> ### Author Response · Authors · 2025-11-21
> **Comment 2/3 for Reviewer 2wEG**
>
> We believe these results support that the value of the adaptive rule lies not in the particular mathematical form of the threshold, but in the **underlying empirical framework** that explains and predicts pruning behavior across LVLMs. We have made this positioning clearer in the revised manuscript.
>
> We appreciate the reviewer’s constructive suggestion, which helped us present the intention and contribution of our method more clearly.
>
> **References:**
>
> [1] Shang, Yuzhang, et al. "Llava-prumerge: Adaptive token reduction for efficient large multimodal models." Proceedings of the IEEE/CVF International Conference on Computer Vision. 2025.
>
> [2] Yang, Senqiao, et al. "Visionzip: Longer is better but not necessary in vision language models." Proceedings of the Computer Vision and Pattern Recognition Conference. 2025.
>
> [3] Zhang, Qizhe, et al. "Beyond text-visual attention: Exploiting visual cues for effective token pruning in vlms." *Proceedings of the IEEE/CVF International Conference on Computer Vision*. 2025.
>
> [4] Long, Sifan, et al. "Beyond attentive tokens: Incorporating token importance and diversity for efficient vision transformers." Proceedings of the IEEE/CVF Conference on Computer Vision and Pattern Recognition. 2023.
>
> [5] Zhang, Qizhe, et al. "Beyond text-visual attention: Exploiting visual cues for effective token pruning in vlms." *Proceedings of the IEEE/CVF International Conference on Computer Vision*. 2025.
>
> ---
>
> **Weakness 2:**  The proposed adaptive thresholding strategy introduces several hyperparameters, notably the scaling coefficients and other implementation choices. These parameters may influence pruning behavior, yet the paper does not provide a clear justification or sensitivity analysis for them.
>
> **Response:**  We sincerely thank the reviewer for pointing out the need to clarify the role of the hyperparameters in our adaptive thresholding strategy. We appreciate the opportunity to explain the original design choices and to present additional analyses conducted during the rebuttal period.
>
> **1. Rationale behind the original parameters.**
>
> In the initial submission, the coefficients (α, β) were not chosen arbitrarily. They were obtained using a simple interpolation based on the **minimum and maximum erank values observed in the LLaVA training set**, mapped to a conservative–aggressive threshold range. This allowed the threshold to scale proportionally with image complexity, providing a smooth transition between attention-focused and diversity-focused pruning regimes.
>
> **2. Sensitivity analysis confirming stability.**
>
> To assess whether these parameters substantially influence model behavior, we conducted a sensitivity study by varying the target threshold range. As summarized below, performance remained highly stable across settings, indicating that the adaptive mechanism does not rely on finely tuned hyperparameters:
>
> | Min Scaling | Max Scaling | MME (Perception) | MME (Cognition) |
> | --- | --- | --- | --- |
> | 0.015 | 0.20 | 1413.3 | 325.3 |
> | 0.015 | 0.25 | 1413.6 | 315.3 |
> | 0.020 | 0.15 | 1418.6 | 326.0 |
> | 0.025 | 0.20 | 1416.4 | 320.7 |
>
> Even with noticeable changes in scaling, the performance fluctuates only minimally. This suggests that the method’s behavior is governed primarily by the **underlying empirical relationship** rather than the particular hyperparameter values.
>
> **3. Simplified statistics-driven formulation (added in the revision).**
>
> To further reduce dependence on explicit coefficients and make the mechanism more transparent, we refined the adaptive thresholding rule in the revised manuscript. The new formulation uses a **dataset-driven ratio**:
>
> $$
> \text{ratio} = \frac{\text{erank}\_{\text{input}}}{\text{erank}\_{\text{avg}}}
> $$
>
> - Complex images (ratio > 1): threshold increases, enabling more aggressive pruning to ensure diversity.
> - Simple images (ratio < 1): threshold decreases, preserving focused and discriminative tokens.
>
> This strategy removes the need for manually tuned parameters and directly operationalizes the empirical finding that image complexity predicts the preferred pruning paradigm.
>
> **4. Validation across diverse LVLM architectures.**
>
> To ensure that this simplified rule is robust and not tied to a specific backbone, we applied it to multiple architectures—**LLaVA-1.5-7B**, **LLaVA-1.5-13B**, **LLaVA-NeXT-7B**, and **Qwen2.5-VL-7B**—during the rebuttal period (**Appendix B.1**). In all cases, the adaptive rule produced consistent improvements over fixed strategies, confirming that its behavior is both stable and architecture-agnostic.
>
> We have revised **Section 4.3** to reflect this improved formulation and added the corresponding results in the appendix. We hope these clarifications address the reviewer’s concern and demonstrate the robustness and principled nature of our adaptive thresholding mechanism.
>
> ---

---

> ### Author Response · Authors · 2025-11-21
> **Comment 3/3 for Reviewer 2wEG**
>
> **Weakness 3:**  Prior token pruning works (e.g., DivPrune, VisionZip, PruMerge+) already explore balancing attention and diversity. This paper’s “adaptive” version appears incremental and lacks a compelling distinction.
>
> **Response:**  We sincerely thank the reviewer for raising this point. We agree that several prior pruning approaches—such as VisionZip [1], and PruMerge+ [2]—combine attention- and diversity-related cues. However, our work differs fundamentally in intention and contribution. Rather than proposing another way to mix the two signals, our goal is to **identify and explain previously unobserved behaviors in LVLM pruning** and to show that these behaviors have concrete implications for existing methods.
>
> In particular, our analysis reveals several findings that, to our knowledge, are *not addressed* by prior works:
>
> 1. **Hybrid methods do not reliably preserve diversity.**
>
>     Although methods like VisionZip [1] and PruMerge+ [2] and  conceptually incorporate diversity, our erank-based measurements show that the token sets they actually retain have surprisingly low erank—often much closer to attention-based pruning than expected. This contradicts the assumption that hybrid pruning naturally balances the two criteria.
>
> 2. **Diversity of the retained tokens strongly correlates with hallucination behavior.**
>
>     Prior pruning works evaluate only accuracy or efficiency. None analyze how pruning affects hallucination, nor discover that high-erank selections tend to increase hallucination while low-erank selections suppress it. This connection has not been explored in LVLM pruning literature.
>
> 3. **The optimal pruning paradigm shifts systematically with image complexity.**
>
>     Existing methods implicitly assume that a single global strategy (attention, diversity, or a fixed mixture) is sufficient. Our results demonstrate that attention-based pruning excels on simple, low-erank images, while diversity-based pruning excels on complex, high-erank images. This image-complexity–dependent behavior is not captured by any prior pruning method.
>
>
> Taken together, these observations indicate that prior works do not fully characterize how attention and diversity interact inside LVLMs; rather, they rely on heuristically combining cues without understanding when each mechanism is appropriate.
>
> Our adaptive rule is therefore not intended as an incremental variation of hybrid pruning. Instead, it provides a minimal operationalization of the empirical principle uncovered above—namely, that pruning effectiveness depends on image complexity. The strength of this principle is further evidenced by the fact that applying our erank-guided rule to existing methods (e.g., BAT [3], VisPruner [4]) during the rebuttal period leads to consistent improvements, while applying the inverse rule consistently degrades performance.
>
> We have emphasized this distinction more clearly in the revised manuscript. We greatly appreciate the reviewer’s comment, which helped us refine the positioning of our contribution.
>
> References:
>
> [1] Yang, Senqiao, et al. "Visionzip: Longer is better but not necessary in vision language models." Proceedings of the Computer Vision and Pattern Recognition Conference. 2025.
>
> [2] Shang, Yuzhang, et al. "Llava-prumerge: Adaptive token reduction for efficient large multimodal models." Proceedings of the IEEE/CVF International Conference on Computer Vision. 2025.
>
> [3] Long, Sifan, et al. "Beyond attentive tokens: Incorporating token importance and diversity for efficient vision transformers." Proceedings of the IEEE/CVF Conference on Computer Vision and Pattern Recognition. 2023.
>
> [4] Zhang, Qizhe, et al. "Beyond text-visual attention: Exploiting visual cues for effective token pruning in vlms." *Proceedings of the IEEE/CVF International Conference on Computer Vision*. 2025.
>
> ---

---

### Official Review · Reviewer_17pH · 2025-11-03

**Soundness:** 3
**Presentation:** 3
**Contribution:** 2
**Rating:** 6
**Confidence:** 2

**Summary:**

This paper presents a comprehensive empirical study on visual token pruning in Large Vision-Language Models (LVLMs), which are computationally expensive due to the large number of visual tokens produced by image encoders. Existing pruning strategies generally fall into two categories:

- Attention-based methods, which retain tokens with high attention scores.

- Diversity-based methods, which keep tokens that are dissimilar to others to preserve feature diversity.

Key Findings:

- Impact of Image Complexity. The authors quantify image complexity using two metrics: Attention entropy for measures how concentrated the model’s attention is. Effective rank (erank) for measures the diversity of token embeddings.

- Hallucination Analysis: Using the CHAIR dataset, the study shows that Attention-based methods produce more conservative outputs with lower hallucination rates. Diversity-based methods generate richer but more hallucinated descriptions due to speculative inclusion of objects.

- Adaptive Pruning Framework: Based on these insights, the authors propose AdaVTP (Adaptive Visual Token Pruning), which dynamically balances attention concentration and token diversity. The method adjusts a similarity threshold (τ) for pruning redundant tokens based on image complexity (measured by erank). A low τ is used for simple images (preserving fine details), while a higher τ enhances diversity for complex images.

**Strengths:**

Comprehensive empirical analysis and validation:
- It goes beyond performance reporting and explores why each method behaves differently, grounded in measurable concepts like attention entropy and effective rank (erank).
- Extensive experiments on nine multimodal benchmarks (VQAv2, GQA, TextVQA, ScienceQA, MMBench, etc.) and the CHAIR hallucination dataset demonstrate the robustness of the approach.

Insightful findings with practical relevance：
- The study reveals clear patterns: attention-based pruning excels in simple images (low erank) while diversity-based pruning performs better in complex ones (high erank).

- These findings can directly guide pruning strategy selection in real-world LVLM deployments where image complexity varies widely.

**Weaknesses:**

Limited novelty in algorithmic design:
- The proposed adaptive pruning framework (AdaVTP) mainly combines two existing ideas — attention-based and diversity-based pruning — using an adaptive threshold determined by image complexity.
- While insightful, this combination strategy is heuristic rather than fundamentally new in algorithmic form.

Limited scope of model diversity:
- Most experiments are based on a single LVLM backbone (LLaVA-1.5-7B).
- The generalizability of the findings to other architectures (e.g., Qwen-VL) is not demonstrated.

Computational trade-off not deeply analyzed:
- The paper claims efficiency improvements but provides limited quantitative analysis on actual latency or memory usage.

Potential instability in adaptive thresholding:
- Since the adaptive rule relies on erank estimated from token embeddings, noisy or atypical embeddings (e.g., from cluttered or low-quality images) might lead to inconsistent threshold adjustments.
- The paper lacks ablation or sensitivity studies showing the stability of AdaVTP under such variations.

**Questions:**

See weakness

---

> ### Author Response · Authors · 2025-11-21
> **Comment 1/4 for Reviewer 17pH**
>
> We thank the reviewer for the careful reading and thoughtful comments. Below, we address the reviewer’s questions and provide detailed explanations. The corresponding revisions in the manuscript are marked in blue for clarity. We hope that the responses below, along with the updated manuscript, address the reviewer’s concerns thoroughly. Due to character limits on OpenReview, our response has been divided into four parts for ease of readability. Thank you for your understanding.
>
> **Weakness 1:** Limited novelty in algorithmic design: The proposed adaptive pruning framework (AdaVTP) mainly combines two existing ideas — attention-based and diversity-based pruning — using an adaptive threshold determined by image complexity. While insightful, this combination strategy is heuristic rather than fundamentally new in algorithmic form.
>
> **Response:**  We appreciate the concern regarding algorithmic novelty, and we have revised the manuscript to clarify the intention and contribution of our adaptive strategy.
>
> Our goal is not to introduce a new architectural mechanism, but rather to provide a **principled and empirically grounded framework** for understanding when attention-driven vs. diversity-driven pruning should be preferred in LVLMs. As detailed in the revised Section 4, our analysis reveals *consistent and previously unreported patterns* linking (i) the diversity of retained token sets, (ii) hallucination behavior, and (iii) image-level complexity measured via erank. The adaptive rule we propose is intentionally simple, functioning as a **direct operationalization of these empirical principles** rather than as a heuristic combination of two existing methods.
>
> Importantly, we demonstrate that the insights derived from our analysis extend beyond our own method. During the rebuttal period, we conducted additional experiments to verify that this empirical principle is not limited to our own pruning strategy. Applying the simple erank-guided linear mapping to existing methods—which originally rely on fixed importance–diversity ratios, such as BAT [1] and VisPruner [2]—consistently improves their performance. In contrast, using the inverse mapping leads to a predictable drop in accuracy across benchmarks. This method-agnostic behavior supports that the contribution lies not in the mechanics of the threshold itself, but in the **framework that explains and predicts pruning behavior across diverse LVLMs**. All corresponding analyses and results have been incorporated into the revised **Section 4.3.**
>
> | Method | Retain tokens | GQA | ScienceQA | POPE | MME | Rel. |
> | --- | --- | --- | --- | --- | --- | --- |
> | **LLaVA-1.5-7B** | 576 | 61.9 | 69.5 | 85.9 | 1862 | 100.0% |
> | BAT (CVPR’23) | 128 | 58.6 | 69.3 | 85.3 | 1737 | 96.75% |
> | BAT + Inverse  | 128 | 58.4 (-0.2) | 69.1 (-0.2) | 85.0 (-0.3) | 1734 (-3) | 96.46% (-0.19) |
> | **BAT + Ours** | 128 | **58.8 (+0.2)** | **69.4 (+0.1)** | **86.8 (+0.9)** | **1782 (+45)** | **97.91% (+1.11)** |
> | Vispruner (ICCV’25) | 128 | 58.2 | 69.0 | 84.6 | 1768 | 96.72% |
> | Vispruner + Inverse  | 128 | 57.9 (-0.3) | 68.8 (-0.2) | 85.3 (+0.7) | 1744 (-24) | 96.35% (-0.37) |
> | **Vispruner + Ours** | 128 | **58.6 (+0.4)** | **69.1 (+0.1)** | **85.5 (+0.9)** | **1787 (+19)** | **97.32% (+0.60)** |
> | BAT (CVPR’23) | 64 | 56.6 | 68.8 | 81.5 | 1683 | 93.91% |
> | BAT + Inverse | 64 | 55.8 (-0.8) | 68.7 (-0.1) | 79.7 (-1.8) | 1602 (-81) | 91.98% (-1.93) |
> | **BAT + Ours** | 64 | **56.9 (+0.3)** | **69.3 (+0.5)** | **83.5 (+2.0)** | **1692 (+9)** | **94.85% (+0.94)** |
> | Vispruner (ICCV’25) | 64 | 55.4 | 69.1 | 80.4 | 1650 | 92.78% |
> | Vispruner + Inverse | 64 | 55.2 (-0.2) | 68.9 (-0.2) | 80.2 (-0.2) | 1643 (-7) | 91.83% (-0.95) |
> | **Vispruner + Ours** | 64 | **55.9 (+0.5)** | **69.3 (+0.2)** | **81.5 (+1.1)** | **1671 (+21)** | **93.76% (+0.98)** |
>
> We also observe the same trend for heterogeneous mixtures (DivPrune + FasterVLM). This combination also benefited from the erank-guided rule, confirming that the discovered principle is method-agnostic and captures general LVLM pruning behavior rather than a heuristic tied to a single model.
>
> | Method | Retain tokens | GQA | ScienceQA | POPE | MME | Rel. |
> | --- | --- | --- | --- | --- | --- | --- |
> | **LLaVA-1.5-7B** | 576 | 61.9 | 69.5 | 85.9 | 1862 | 100.0% |
> | Divprune (CVPR’25)   | 128 | 59.4 | 68.6 | 87.0 | 1707 | 96.90% |
> | FasterVLM (ArXiv’24) | 128 | 57.85 | 68.47 | 83.2 | 1757 | 95.80% |
> | Divprune+FasterVLM (fixed)  | 128 | 58.35 | 68.52 | 84.9 | 1768 | 96.66% |
> | **Divprune+FasterVLM (adaptive)** | **128** | **58.87** | **68.82** | **85.99** | **1787** | **97.55%** |
> | Divprune (CVPR’25)   | 64 | 57.5 | 68.0 | 85.5 | 1615 | 94.25% |
> | FasterVLM   (ArXiv’24) | 64 | 55.03 | 69.0 | 77.4 | 1665 | 91.91% |
> | Divprune+FasterVLM (fixed) | 64 | 56.75 | 68.57 | 82.2 | 1681 | 94.08% |
> | **Divprune+FasterVLM (adaptive)** | **64** | **57.23** | **68.86** | **83.5** | **1690** | **94.87%** |

---

> ### Author Response · Authors · 2025-11-21
> **Comment 2/4 for Reviewer 17pH**
>
> To avoid overstating the contribution as algorithmic in nature, we have revised the manuscript to emphasize that our study aims to establish a **generalizable empirical principle** for LVLM pruning and to show how a simple thresholding rule derived from that principle can improve different pruning strategies. We hope this clarification makes the intended novelty—grounded in analysis rather than architecture—more explicit.
>
> **Reference:**
>
> [1] Long, Sifan, et al. "Beyond attentive tokens: Incorporating token importance and diversity for efficient vision transformers." Proceedings of the IEEE/CVF Conference on Computer Vision and Pattern Recognition. 2023.
>
> [2] Zhang, Qizhe, et al. "Beyond text-visual attention: Exploiting visual cues for effective token pruning in vlms." *Proceedings of the IEEE/CVF International Conference on Computer Vision*. 2025.
>
> ---
>
> **Weakness 2:** Limited scope of model diversity: Most experiments are based on a single LVLM backbone (LLaVA-1.5-7B). The generalizability of the findings to other architectures (e.g., Qwen-VL) is not demonstrated.
>
> **Response:**  We sincerely thank the reviewer for raising this important point regarding model diversity and generalizability. We fully agree that demonstrating the robustness of our findings across different LVLM architectures is essential. In response to this valuable suggestion, we have significantly expanded the scope of our experiments in the revised version.
>
> First, to ensure scalability within the LLaVA family, we additionally evaluate our method on **LLaVA-1.5-13B** [1] and **LLaVA-NeXT-7B**. [2] These results are provided in the **Appendix  B.1**. Across both larger and more recent LLaVA variants, we observe the same trends discussed in the main paper: attention-focused pruning remains advantageous on low-erank images, while diversity-focused selection performs better on complex, high-erank images. Our adaptive strategy consistently improves performance across all tested model sizes.
>
> Second, embracing the reviewer’s suggestion to validate generalizability beyond the LLaVA architecture, we conducted new experiments on **Qwen2.5-VL-7B [3]**. This model differs substantially in both vision encoder and multimodal alignment design, making it a strong test for cross-architecture robustness. Again, we find that our erank-based characterization and adaptive rule hold consistently. The same image-complexity pattern emerges, and applying our adaptive threshold yields clear improvement over fixed-strategy pruning. The extended results on Qwen2.5-VL-7B have been newly included in **Appendix B.1.**
>
> Representative results:
>
> | Method | Retain Tokens | TextVQA | ChartQA  | AI2D  | OCRBench  | MME (Perc.)  | MME (Cogn.) | MMB-EN | MMB-CN | Rel. |
> | --- | --- | --- | --- | --- | --- | --- | --- | --- | --- | --- |
> | Qwen2.5-VL-7B | 1225 | 82.1 | 77.5 | 83.0 | 84.1 | 1691 | 642 | 83.2 | 83.2 | 100.0% |
> | DivPrune (CVPR'25) | 512 | 79.7 | 72.0 | 81.9 | 78.5 | 1704 | 620 | 81.8 | 81.0 | 96.8% |
> | **Ours** | 512 | 79.8 | 71.5 | 82.1 | 78.5 | 1700 | 630 | 81.3 | 80.8 | **96.9%** |
> | DivPrune (CVPR'25) | 256 | 75.2 | 62.9 | 79.7 | 66.4 | 1703 | 542 | 80.3 | 79.8 | 90.3% |
> | **Ours** | 256 | 74.2 | 62.4 | 80.5 | 65.8 | 1670 | 630 | 80.9 | 80.3 | **92.1%** |
>
> Across all architectures, our adaptive strategy follows a simple and unified principle:
>
> - for simple images (low erank), the threshold becomes stricter to preserve focused, high-attention evidence;
> - for complex images (high erank), a looser threshold encourages diverse token retention to capture distributed semantics.
>
> This mechanism remains unchanged and continues to produce stable gains regardless of model size or architecture. We believe these additions demonstrate that our findings are not tied to a single backbone, and that the empirical principles uncovered in our analysis are robust, model-agnostic, and broadly applicable across LVLM architectures. We appreciate the reviewer’s insightful suggestion, which has allowed us to further strengthen the generality of our work.
>
> **References:**
>
> [1] Liu, Haotian, et al. "Improved baselines with visual instruction tuning." *Proceedings of the IEEE/CVF conference on computer vision and pattern recognition*. 2024.
>
> [2] Li, Bo; Zhang, Kaichen; Zhang, Hao; Guo, Dong; Zhang, Renrui; Li, Feng; Zhang, Yuanhan; Liu, Ziwei; Li, Chunyuan. *“LLaVA-NeXT: Stronger LLMs Supercharge Multimodal Capabilities in the Wild.”* 10 May 2024. URL: [https://llava-vl.github.io/blog/2024-05-10-llava-next-stronger-llms/](https://llava-vl.github.io/blog/2024-05-10-llava-next-stronger-llms/?utm_source=chatgpt.com)
>
> [3] Bai, Shuai, et al. "Qwen2. 5-vl technical report." *arXiv preprint arXiv:2502.13923* (2025).
>
> ---

---

> ### Author Response · Authors · 2025-11-21
> **Comment 3/4 for Reviewer 17pH**
>
> **Weakness 3:** Computational trade-off not deeply analyzed: The paper claims efficiency improvements but provides limited quantitative analysis on actual latency or memory usage.
>
> **Response:**  We thank the reviewer for highlighting the need for a deeper analysis of computational trade-offs. We agree that the initial submission did not sufficiently quantify efficiency beyond FLOPs, and we have substantially expanded our evaluation in the revised manuscript.
>
> To address this concern, we now report latency, GPU memory usage, and practical runtime measured on RTX 4090 at TextVQA [1] dataset, in addition to FLOPs. We further include comparisons with both pre-pruning methods and recent in-LLM pruning approaches such as PDrop (CVPR’25) [2] and SparseVLM (ICML’25) [3].
>
> A summary of the extended efficiency analysis is provided below (full details appear in **Appendix A**):
>
> | Method | Retain | FLOPs (T) | Latency (ms/sample) | GPU Mem (GB) | Acc (%) |
> | --- | --- | --- | --- | --- | --- |
> | Vanilla (LLaVA-1.5-7B) | 576 | 3.14 | 172 | 13.60 | 58.2 |
> | PDrop (CVPR’25) | 64 | 0.51 | 128 | 13.30 | 55.0 |
> | SparseVLM (ICML’25) | 64 | 0.52 | 129 | 16.26 | 55.2 |
> | DivPrune (CVPR’25) | 64 | 0.48 | 110 | 13.30 | 55.8 |
> | VisPruner (ICCV’25) | 64 | 0.48 | 115 | 13.30 | 55.4 |
> | **Ours** | 64 | 0.48 | 115 | 13.30 | 56.0 |
>
> Our method reduces FLOPs by 89.0% (3.14T → 0.48T) while maintaining 96.2% of the original accuracy. Importantly, because visual tokens are removed before entering the LLM, every subsequent decoder layer benefits from the reduced sequence length. On real hardware, this yields a 33% latency reduction (172 ms → 115 ms) and a modest but consistent reduction in GPU memory usage.
>
> We also clarify the conceptual difference between pre-pruning (our approach, DivPrune [4], VisPruner [5]) and in-LLM pruning (PDrop, SparseVLM). Although in-LLM methods reduce FLOPs theoretically, they perform pruning inside the decoder, meaning early Transformer layers still process all 576 tokens. As a result, end-to-end runtime does not reflect the full FLOPs reduction. For example, SparseVLM achieves similar FLOPs to ours but still exhibits higher latency (129 ms vs. our 115 ms) and significantly higher memory usage due to additional text-embedding storage.
>
> In contrast, our pre-pruning approach reduces the sequence length before the LLM, so all layers benefit uniformly from the shorter sequence, and—because tokens are removed upfront—also achieves lower GPU memory usage compared to in-LLM pruning methods. Under identical FLOPs, our method also yields the highest accuracy among pre-pruning baselines.
>
> In addition, our method is compatible with FlashAttention [6], allowing further efficiency gains when combined with modern acceleration techniques.
>
> We have incorporated the full extended analysis—including latency profiling, GPU memory, and discussion of architectural implications—into **Appendix A**, along with a detailed quantification of the computational overhead introduced by the erank computation. We sincerely appreciate the reviewer’s suggestion, which helped strengthen and clarify the practical efficiency benefits of our approach.
>
> **Reference:**
>
> [1] Singh, Amanpreet, et al. "Towards vqa models that can read." *Proceedings of the IEEE/CVF conference on computer vision and pattern recognition*. 2019.
>
> [2] Zhang, Yuan, et al. "Sparsevlm: Visual token sparsification for efficient vision-language model inference." *arXiv preprint arXiv:2410.04417* (2024).
>
> [3] Xing, Long, et al. "Pyramiddrop: Accelerating your large vision-language models via pyramid visual redundancy reduction." *arXiv preprint arXiv:2410.17247* (2024).
>
> [4] Alvar, Saeed Ranjbar, et al. "Divprune: Diversity-based visual token pruning for large multimodal models." *Proceedings of the Computer Vision and Pattern Recognition Conference*. 2025.
>
> [5] Zhang, Qizhe, et al. "Beyond text-visual attention: Exploiting visual cues for effective token pruning in vlms." *Proceedings of the IEEE/CVF International Conference on Computer Vision*. 2025.
>
> [6] Dao, Tri, et al. "Flashattention: Fast and memory-efficient exact attention with io-awareness." Advances in neural information processing systems 35 (2022): 16344-16359.
>
> ---

---

> ### Author Response · Authors · 2025-11-21
> **Comment 4/4 for Reviewer 17pH**
>
> **Weakness 4:** Potential instability in adaptive thresholding: Since the adaptive rule relies on erank estimated from token embeddings, noisy or atypical embeddings (e.g., from cluttered or low-quality images) might lead to inconsistent threshold adjustments. The paper lacks ablation or sensitivity studies showing the stability of AdaVTP under such variations.
>
> **Response:**
>
> We thank the reviewer for raising this important point about the stability of the adaptive thresholding mechanism. To directly address this concern, we conducted an additional sensitivity analysis following the corruption protocol used in COCO-C [1], a standard benchmark for robustness evaluation. Specifically, we applied 15 corruption types—including noise, blur, weather effects, contrast changes, and compression artifacts—to all 2,374 images in the MME dataset [2] and measured how the resulting erank values deviated from those of the clean images.
>
> | **Corruption Type** | **Mean erank diff. (S=1)** | **Relative change (S=1)** | **Mean erank diff. (S=3)** | **Relative change (S=3)** |
> | --- | --- | --- | --- | --- |
> | Zoom blur | 5.85 | 6.14% | 7.62 | 8.12% |
> | Brightness | 1.48 | 1.56% | 2.52 | 2.70% |
> | Contrast | 2.38 | 2.54% | 4.38 | 4.62% |
> | Defocus blur | 2.06 | 2.28% | 3.74 | 4.16% |
> | Elastic transform | 2.18 | 2.54% | 3.38 | 4.05% |
> | Fog | 3.24 | 3.73% | 4.05 | 4.71% |
> | Frost | 3.96 | 4.56% | 6.10 | 7.24% |
> | Gaussian noise | 2.83 | 3.23% | 4.17 | 4.73% |
> | Impulse noise | 3.34 | 4.04% | 4.37 | 5.10% |
> | JPEG compression | 2.31 | 2.48% | 2.77 | 2.98% |
> | Motion blur | 2.11 | 2.33% | 3.38 | 3.91% |
> | Pixelate | 1.22 | 1.32% | 1.61 | 1.76% |
> | Shot noise | 2.74 | 3.14% | 4.06 | 4.62% |
> | Snow | 3.84 | 4.17% | 5.37 | 5.76% |
> | Glasses blur | 2.15 | 2.46% | 4.07 | 4.79% |
> | **Average** | **2.78** | **3.10%** | **4.11** | **4.62%** |
>
> Across all corruption types, erank remained highly stable. At corruption severity 1, the mean absolute deviation was 2.78, and at severity 3, it was 4.11—corresponding to only 2–4% of the natural erank scale (clean-image mean: 94.86; std: 19.46).
>
> We further observed consistent and interpretable patterns:
>
> - Corruptions that affect global spatial structure (e.g., zoom blur, frost, snow) produced slightly larger deviations (typically 5–7 points), which is expected because they alter higher-level feature distributions.
> - Corruptions that modify only local pixel statistics (e.g., brightness, pixelation, JPEG compression) had minimal impact, generally within 1–2.5 points, since they preserve the global structure of the embedding matrix.
> - Increasing severity from 1 to 3 produced only modest changes in deviation, indicating that erank responds smoothly and proportionally rather than erratically.
>
> We believe this robustness stems from erank’s design: because it reflects the entropy of the singular-value spectrum, it captures global characteristics of the embedding distribution and is therefore naturally less sensitive to localized noise or pixel-level distortion.
>
> Overall, these results suggest that erank-based adaptive thresholding remains stable even under noisy, cluttered, or degraded inputs, and that the adaptive mechanism behaves consistently across a wide range of real-world corruptions. We have included the full robustness tables, additional corruptions, and supporting visualizations in the revised **Appendix E.**
>
> **References:**
>
> [1] Hendrycks, Dan, and Thomas G. Dietterich. "Benchmarking neural network robustness to common corruptions and surface variations." *arXiv preprint arXiv:1807.01697* (2018).
>
> [2] Fu, Chaoyou, et al. "Mme: A comprehensive evaluation benchmark for multimodal large language models." *The Thirty-ninth Annual Conference on Neural Information Processing Systems Datasets and Benchmarks Track*. 2025.
>
> ---

---

> ### Comment · Reviewer_17pH · 2025-11-25
> **Reply to Authors**
>
> Thanks for the author's reply, which has resolved most of my questions. I have raised my score to 8. Good luck!

---

> > ### Author Response · Authors · 2025-11-25
> >
> > Thank you for the update and for taking the time to reconsider the submission.
> > We appreciate your careful evaluation.

---

### Official Review · Reviewer_zvYn · 2025-11-03

**Soundness:** 2
**Presentation:** 3
**Contribution:** 2
**Rating:** 2
**Confidence:** 3

**Summary:**

This paper conducts an empirical analysis of attention-based and diversity-based visual token pruning strategies in vision-language models (VLMs). The study reveals that attention-based pruning strategy shows good performance on simple images (low erank), while diversity-based pruning strategy shows better performance on complex images. It proposes a method that adaptively determines the visual tokens preserved based on the erank of the images.

**Strengths:**

1. Provides diverse experiments to validate its approach
2. The paper is well-written and easy to follow.

**Weaknesses:**

1. I am not sure the findings of the paper is novel enough. [1] shows that "a satisfied pruning method should jointly take the token importance and diversity into account." to preserve both local (important) and global (diverse) information, which is what the paper proposes to do.
2. I think an important related work is missed [2]. It also determines pruning threshold based on input instance adaptively.
3. From my understanding, the number of visual tokens is fixed in the experiments. Why didn’t the authors extend their method to adaptively determine the number of visual tokens retained? It seems that dynamically adjusting the number of preserved tokens could further improve the efficiency and effectiveness of visual token pruning.

[1] Long, Sifan, et al. "Beyond attentive tokens: Incorporating token importance and diversity for efficient vision transformers." Proceedings of the IEEE/CVF Conference on Computer Vision and Pattern Recognition. 2023.

[2] Ye, Xubing, et al. "Atp-llava: Adaptive token pruning for large vision language models." Proceedings of the Computer Vision and Pattern Recognition Conference. 2025.

**Questions:**

Please address questions in the weaknesses section.

---

> ### Author Response · Authors · 2025-11-21
> **Comment 1/3 for Reviewer zvYn**
>
> We thank the reviewer for the careful reading and thoughtful comments. Below, we address the reviewer’s questions and provide detailed explanations. The corresponding revisions in the manuscript are marked in blue for clarity. We hope that the responses below, along with the updated manuscript, address the reviewer’s concerns thoroughly. Due to character limits on OpenReview, our response has been divided into three parts for ease of readability. Thank you for your understanding.
>
> **Weakness 1:** I am not sure the findings of the paper is novel enough. [1] shows that "a satisfied pruning method should jointly take the token importance and diversity into account." to preserve both local (important) and global (diverse) information, which is what the paper proposes to do.
>
> **Response:**  We sincerely thank the reviewer for raising this concern about novelty. We agree that Long et al. [1] (BAT) clearly state that both attention and diversity can be beneficial. However, our goal is not merely to restate this intuition or to combine two heuristics. Instead, our contribution is to provide a **quantitative and empirically validated framework** that:
>
> 1. characterizes how existing LVLM pruning methods trade off attention and diversity by measuring the *diversity* of their retained token sets via effective rank (erank), and linking this to their hallucination behavior; and
> 2. shows that the relative advantage of attention-based vs. diversity-based pruning depends systematically on **image-level complexity**, which naturally leads to a simple erank-guided adaptive rule that improves existing pruning methods (including BAT and VisPruner [2]), not just our own.
>
> **Section 4.1 (Empirical Analysis of Attention-Based and Diversity-Based Pruning).**
>
> In the revised Section 4.1, we characterize existing LVLM pruning methods by **quantifying the diversity of the visual tokens they retain** using effective rank (erank). Unlike [1], which discusses attention and diversity conceptually, we explicitly measure, for each pruning method:
>
> - the erank of its retained visual tokens; and
> - how this retained-token diversity correlates with hallucination rates.
>
> This analysis reveals clear and previously unreported patterns:
>
> - attention-based methods (e.g., FasterVLM [3]) tend to select **low-erank** token sets and exhibit more conservative, low-hallucination behavior;
> - diversity-based methods (e.g., DivPrune [4]) produce **high-erank** token sets and show higher recall but also higher hallucination;
> - hybrid methods (PruMerge+ [5], VisionZip [6], VisPruner [4]) lie in between and, perhaps surprisingly, often do **not** reach the erank levels of purely diversity-oriented pruning, indicating that their retained tokens may still lack sufficient diversity.
>
> As an illustrative example (POPE dataset):
>
> | method | PruMerge+ | VisionZip | VisPruner | DivPrune |
> | --- | --- | --- | --- | --- |
> | **mean erank of retained tokens** | 10.91 | 14.02 | 14.35 | **21.84** |
>
> To our knowledge, prior LVLM pruning works—including BAT, DivPrune, VisionZip, and PruMerge+—have not provided such a **method-by-method quantitative comparison** of the diversity of the actually retained tokens, nor connected it directly to hallucination tendencies. We believe this “erank-as-diversity” interpretation provides a new and practical diagnostic tool for understanding attention–diversity trade-offs in pruning.
>
> **Section 4.2 (Image-complexity–dependent behavior and an erank-guided adaptive rule).**
>
> Building on this characterization of existing methods, the revised Section 4.2 examines how pruning behavior changes with image complexity. Using erank and attention statistics computed over all visual tokens of an input image, we observe that:
>
> - for simple, low-erank images, attention-focused pruning performs better;
> - for complex, high-erank images, diversity-focused pruning becomes more effective.
>
> This perspective also explains why certain pruning methods excel on particular datasets (e.g., ScienceQA vs. POPE vs. MME), and to the best of our knowledge, this image-complexity view of pruning effectiveness is new in LVLMs.

---

> ### Author Response · Authors · 2025-11-21
> **Comment 2/3 for Reviewer zvYn**
>
> **Method-agnostic validation.**
>
> To further validate that our findings are general rather than tied to our own algorithm, we additionally conducted experiments during the rebuttal period in which we applied the simple erank-guided linear mapping to existing methods—which originally rely on fixed importance–diversity ratios, such as BAT (as the reviewr noted) [1] and VisPruner [2]—consistently improves their performance.
>
> Across both 128- and 64-token budgets:
>
> - applying our adaptive rule consistently **improves performance**;
> - applying the *inverse* rule (which contradicts the discovered image-complexity trend) consistently **reduces performance**.
>
> Representative results (All corresponding analyses and results have been incorporated into the revised **Section 4.3.)**:
>
> | Method | Retain tokens | GQA | ScienceQA | POPE | MME | Rel.  |
> | --- | --- | --- | --- | --- | --- | --- |
> | **LLaVA-1.5-7B** | 576 | 61.9 | 69.5 | 85.9 | 1862 | 100.0% |
> | BAT (CVPR’23) | 128 | 58.6 | 69.3 | 85.3 | 1737 | 96.75% |
> | BAT + Inverse  | 128 | 58.4 (-0.2) | 69.1 (-0.2) | 85.0 (-0.3) | 1734 (-3) | 96.46% (-0.19) |
> | **BAT + Ours** | 128 | **58.8 (+0.2)** | **69.4 (+0.1)** | **86.8 (+0.9)** | **1782 (+45)** | **97.91% (+1.11)** |
> | Vispruner (ICCV’25) | 128 | 58.2 | 69.0 | 84.6 | 1768 | 96.72% |
> | Vispruner + Inverse  | 128 | 57.9 (-0.3) | 68.8 (-0.2) | 85.3 (+0.7) | 1744 (-24) | 96.35% (-0.37) |
> | **Vispruner + Ours** | 128 | **58.6 (+0.4)** | **69.1 (+0.1)** | **85.5 (+0.9)** | **1787 (+19)** | **97.32% (+0.60)** |
> | BAT (CVPR’23) | 64 | 56.6 | 68.8 | 81.5 | 1683 | 93.91% |
> | BAT + Inverse | 64 | 55.8 (-0.8) | 68.7 (-0.1) | 79.7 (-1.8) | 1602 (-81) | 91.98% (-1.93) |
> | **BAT + Ours** | 64 | **56.9 (+0.3)** | **69.3 (+0.5)** | **83.5 (+2.0)** | **1692 (+9)** | **94.85% (+0.94)** |
> | Vispruner (ICCV’25) | 64 | 55.4 | 69.1 | 80.4 | 1650 | 92.78% |
> | Vispruner + Inverse | 64 | 55.2 (-0.2) | 68.9 (-0.2) | 80.2 (-0.2) | 1643 (-7) | 91.83% (-0.95) |
> | **Vispruner + Ours** | 64 | **55.9 (+0.5)** | **69.3 (+0.2)** | **81.5 (+1.1)** | **1671 (+21)** | **93.76% (+0.98)** |
>
> We also observe the same trend for heterogeneous mixtures (DivPrune + FasterVLM). this combination also benefited from the erank-guided rule, confirming that the discovered principle is method-agnostic and captures general LVLM pruning behavior rather than a heuristic tied to a single model. These additional results have been incorporated into the revised **Section 4.3.**
>
> Representative results:
>
> | Method | Retain tokens | GQA | ScienceQA | POPE | MME | Rel. |
> | --- | --- | --- | --- | --- | --- | --- |
> | **LLaVA-1.5-7B** | 576 | 61.9 | 69.5 | 85.9 | 1862 | 100.0% |
> | Divprune (CVPR’25)   | 128 | 59.4 | 68.6 | 87.0 | 1707 | 96.90% |
> | FasterVLM (ArXiv’24) | 128 | 57.85 | 68.47 | 83.2 | 1757 | 95.80% |
> | Divprune+FasterVLM (fixed)  | 128 | 58.35 | 68.52 | 84.9 | 1768 | 96.66% |
> | **Divprune+FasterVLM (adaptive)** | **128** | **58.87** | **68.82** | **85.99** | **1787** | **97.55%** |
> | Divprune (CVPR’25)   | 64 | 57.5 | 68.0 | 85.5 | 1615 | 94.25% |
> | FasterVLM   (ArXiv’24) | 64 | 55.03 | 69.0 | 77.4 | 1665 | 91.91% |
> | Divprune+FasterVLM (fixed) | 64 | 56.75 | 68.57 | 82.2 | 1681 | 94.08% |
> | **Divprune+FasterVLM (adaptive)** | **64** | **57.23** | **68.86** | **83.5** | **1690** | **94.87%** |
>
>
> We appreciate the reviewer’s pointer to [1]. The revised Section 4 clarifies that our contribution is not simply about combining attention and diversity. Rather, we provide: (i) a **quantitative erank-based framework** for understanding what current pruning methods actually retain, (ii) evidence that **image complexity governs** when attention vs. diversity is preferable, and (iii) a **simple erank-guided rule** that consistently improves multiple existing pruning approaches.
>
> **References:**
>
> [1] Long, Sifan, et al. "Beyond attentive tokens: Incorporating token importance and diversity for efficient vision transformers." Proceedings of the IEEE/CVF Conference on Computer Vision and Pattern Recognition. 2023.
>
> [2] Zhang, Qizhe, et al. "Beyond text-visual attention: Exploiting visual cues for effective token pruning in vlms." *Proceedings of the IEEE/CVF International Conference on Computer Vision*. 2025.
>
> [3] Zhang, Qizhe, et al. "[CLS] Attention is All You Need for Training-Free Visual Token Pruning: Make VLM Inference Faster." *arXiv e-prints* (2024): arXiv-2412.
>
> [4] Alvar, Saeed Ranjbar, et al. "Divprune: Diversity-based visual token pruning for large multimodal models." *Proceedings of the Computer Vision and Pattern Recognition Conference*. 2025.
>
> [5] Shang, Yuzhang, et al. "Llava-prumerge: Adaptive token reduction for efficient large multimodal models." Proceedings of the IEEE/CVF International Conference on Computer Vision. 2025.
>
> [6] Yang, Senqiao, et al. "Visionzip: Longer is better but not necessary in vision language models." Proceedings of the Computer Vision and Pattern Recognition Conference. 2025.

---

> ### Author Response · Authors · 2025-11-21
> **Comment 3/3 for Reviewer zvYn**
>
> **Weakness 2, 3:** I think an important related work is missed [2]. It also determines pruning threshold based on input instance adaptively. From my understanding, the number of visual tokens is fixed in the experiments. Why didn’t the authors extend their method to adaptively determine the number of visual tokens retained? It seems that dynamically adjusting the number of preserved tokens could further improve the efficiency and effectiveness of visual token pruning.
>
> **Response:**
>
> We thank the reviewer for highlighting this relevant work. We agree that **ATP-LLaVA [1]** is an important instance-wise adaptive pruning method, and we now explicitly cite and discuss it in the revised **Related Work section.** ATP-LLaVA predicts the pruning ratio for each input using a lightweight controller inside the LLM decoder, with the main goal of improving computational efficiency.
>
> Our notion of “adaptive,” however, is motivated differently and complements. As clarified in the revised Section 4, our work analyzes how attention-based vs. diversity-based pruning behave under different image conditions, using effective rank to quantify the diversity of retained tokens and to explain why these paradigms perform differently across inputs. Based on consistent empirical trends, we propose a simple erank-guided rule that adaptively balances attention-based and diversity-based selection *per input*, while keeping the token budget fixed.
>
> We chose fixed budgets in our main experiments to isolate the effect of which tokens are selected (attention vs. diversity) and to ensure fair comparisons with prior pruning methods, which also assume fixed budgets.
>
> Following the reviewer’s suggestion, we additionally conducted an experiment in which the number of retained visual tokens is also adapted based on the image’s effective rank. Starting from a reference budget of 88 tokens, low-erank (simple) images reduce the count by up to 20%, while high-erank (complex) images proportionally increase it. This reflects our observation that lower-erank images have more concentrated visual information and can be represented with fewer tokens, whereas higher-erank images exhibit more dispersed information and therefore benefit from more tokens. Since this erank-based adjustment does not preserve an exact count across samples, we report the average number of retained tokens per dataset.
>
> | Method | Retain tokens | GQA | SQA | POPE | MME | MMBench | Rel. |
> | --- | --- | --- | --- | --- | --- | --- | --- |
> | LLaVA-1.5-7B | 576 | 62 | 71.6 | 85.8 | 1510 | 64.3 | 100% |
> | ATP-LLaVA | 88 (average) | 56.8 | 67.2 | 82.8 | 1401 | 64.7 | 95.07% |
> | Proposed (fixed count) | 88 | 58.1 | 68.0 | 84.2 | 1405 | 62.3 | 95.35% |
> | **Proposed + Adaptive Count** | 85.5 (average) | 58.36 (86.2) | 68.2 (82.1) | 85.2 (90.8) | 1408 (84.2) | 63.1 (84.2) | **96.01%** |
>
>
> Overall, the adaptive-count configuration achieves **generally accuracy improvements without any additional training while also providing slight additional efficiency gains**, making it more effective than using a fixed token count, consistent with the reviewer’s intuition. We will emphasize that this direction is complementary to ATP-LLaVA and represents a promising extension of our erank-guided framework. We have included extended results for this adaptive-count variant in **Appendix B.5.**
>
> **References:**
>
> [1] Ye, Xubing, et al. "Atp-llava: Adaptive token pruning for large vision language models." Proceedings of the Computer Vision and Pattern Recognition Conference. 2025.

---

### Author Response · Authors · 2025-11-21
**Summary of Revision**

# **Summary of Revisions**

We thank all reviewers for their constructive feedback. Below we summarize the major updates made in the revised manuscript.

---

## **Key Updates**

- **Parameter-Free Adaptive Rule:**

    Replaced the initial (α, β) coefficients with a **fully statistics-driven, parameter-free** rule based on the erank ratio between the input image and dataset average.

- **Method-Agnostic Improvements:**

    Verified that our **erank-guided adaptive principle** consistently improves multiple pruning methods (BAT, VisPruner, DivPrune+FasterVLM), while the inverse mapping reliably degrades performance.

- **Unified Diversity Measurement:**

    Enabled a **consistent erank-based analysis** to quantify how well existing pruning methods preserve token-set diversity, allowing direct comparison of their actual retained-token diversity behaviors.

---

## **Main Paper Revisions**

- **Updated Adaptive Rule Description:**

    Parameter-free threshold formulation and its motivation. in **Sec 4.3.**

- **Integrated Additional Experiments:**

    Added new analyses demonstrating method-agnostic improvements and expanded discussion on pruning behavior across image types in **Sec 4.3**.

- **Erank-Based Analysis of Existing Pruning Methods:**

    Our contribution is to analyze existing diversity-based and attention-based pruning methods through erank, and to uncover the image-complexity–dependent behaviors that naturally emerge from this analysis in **Sec 4.1**.

---

## **Appendix Updates**

- **Runtime & Efficiency Analysis :**

    Added detailed breakdowns of erank overhead, latency, FLOPs, and GPU memory, including comparisons with in-LLM pruning methods in **Appendix A**.

- **Generalization Across LVLMs:**

    Added evaluations on **Qwen2.5-VL-7B** and additional LLaVA variants, confirming architecture-agnostic behavior in **Appendix B.1**.

- **Qualitative Reasoning Examples:**

    Provided detailed case studies on counting, spatial reasoning, and existence tasks to illustrate strengths and failure modes in **Appendix F.**

- **Robustness & Sensitivity Studies:**

    Included the measurement results for the robustness of erank, using the COCO-C corruption dataset, in **Appendix E**.

- **Attention entropy & erank Scale Analysis:** distribution statistics and histograms clarifying the natural scale of attention entropy and its relationship to eranK in **Appendix D.**
- **Entropy-Based Adaptation:** experiments showing that entropy-based adaptation achieves trends similar to erank-based adaptation **(Appendix B.2).**

- **Dynamic Token pruning:** Expanded our analysis to show that complex images benefit from retaining more tokens, while simple images tolerate more aggressive pruning. **We achieves higher accuracy and efficiency than both fixed-token configurations and prior dynamic pruning methods (Appendix B.5).**

---
## **Key Contributions**

- **Empirical Framework:**

    A principled erank-based perspective that explains when attention-based or diversity-based pruning is preferable.

- **Method-agnositc Adaptive Rule:** A simple mechanism derived from our empirical findings, which consistently improves diverse pruning methods (e.g., BAT, VisPruner) without requiring architectural changes.
- **Quantifying Token Diversity via erank:** A quantitative erank-based evaluation of how much feature diversity is maintained across pruning strategies.

We believe these revisions address the reviewers’ concerns and strengthen the paper’s contributions. Thank you for your thoughtful feedback!

---

### Author Response · Authors · 2025-12-01
**Final AC Summary (2/2)**

## **3. Specific Concerns**

### **A. Why not vary the number of retained tokens?**

*(zvYn = 2)*

- Explained rationale: fixed budgets allow fair comparison with prior pruning work
- Added an **adaptive-count experiment** showing:
    - Accuracy & efficiency improvements
    - Outperforms *ATP-LLaVA, as mentioned by the zvYn,* under comparable conditions **(Appendix B.5)**

    **→ No reply before interruption**


---

### **B. Robustness of erank under corrupted inputs**

*(17pH = 6→8)*

- Full COCO-C corruption analysis shows erank varies only 2–4% **(Appendix E)**

    **→ Resolved**


---

### **C. Statistical significance**

*(DUt4 = 4→6)*

- All experiments use **temperature = 0**, eliminating stochasticity

    **→ Resolved**


---

### **D. Entropy Scale Clarification**

*(g8h8 = 8)*

- Added entropy/erank histograms to clarify their scale. **(Appendix D)**

    **→ No reply before interruption**


---

### **E. Failures in fine-grained reasoning**

*(DUt4 = 4→6)*

- Initial qualitative analysis was insufficient. (Appendix F.1)

→ We expanded this as requested

**Final update added in Appendix F.2:**

- Two clear failure modes identified:
    - **Low-erank images with many dispersed objects**

        → behaves too similar to attention-based pruning

    - **High-erank images with localized cues**

        → behaves too similar to diversity-based pruning


    This provides the systematic understanding the reviewer asked for.


**→ No follow-up due to early interruption**

---

We provide this summary to ensure the constructive developments during the discussion—prior to the platform interruption—are fully visible to the Area Chairs.

We sincerely appreciate your effort and thoughtful assessment under these unusual circumstances.



**Best regards,**

**The Authors**

---

**References:**

[1] Long, Sifan, et al. "Beyond attentive tokens: Incorporating token importance and diversity for efficient vision transformers." Proceedings of the IEEE/CVF Conference on Computer Vision and Pattern Recognition. 2023.

[2] Zhang, Qizhe, et al. "Beyond text-visual attention: Exploiting visual cues for effective token pruning in vlms." *Proceedings of the IEEE/CVF International Conference on Computer Vision*. 2025.

[3] Alvar, Saeed Ranjbar, et al. "Divprune: Diversity-based visual token pruning for large multimodal models." *Proceedings of the Computer Vision and Pattern Recognition Conference*. 2025.

[4] Zhang, Qizhe, et al. "[CLS] Attention is All You Need for Training-Free Visual Token Pruning: Make VLM Inference Faster." *arXiv e-prints* (2024): arXiv-2412.

[5] Lin, Zhihang, et al. "Boosting multimodal large language models with visual tokens withdrawal for rapid inference." *Proceedings of the AAAI Conference on Artificial Intelligence*. Vol. 39. No. 5. 2025.

[6] Yang, Senqiao, et al. "Visionzip: Longer is better but not necessary in vision language models." Proceedings of the Computer Vision and Pattern Recognition Conference. 2025.

---

---

### Author Response · Authors · 2025-12-01
**Final AC Summary (1/2)**

### **Dear Area Chair,**

We sincerely appreciate your effort in managing the unexpected OpenReview rollback and ensuring a fair evaluation process.

Since all discussion messages were reverted to their pre-rebuttal state, we would like to provide an accurate and concise summary of the progress made during the discussion phase **prior to the interruption on Nov. 27**.

---

## **1. Score Changes Before the Rollback**

During the discussion (Nov. 21–27), **two reviewers updated their scores upward** after reviewing our clarifications and revisions:

- **Reviewer 17pH:** 6 → **8** (Nov. 25)

    *“Thanks for the author's reply, which has resolved most of my questions.”*

- **Reviewer DUt4:** 4 → **6** (Nov. 21)

    *“The clarifications meaningfully reduce my earlier concerns.”*


The remaining reviewers (**zvYn = 2, 2wEG = 2, g8h8 = 8**) did not have a chance to respond before the system shutdown.

Thus, prior to the rollback, the score distribution improved from:

**[8, 6, 4, 2, 2] → [8, 8, 6, 2, 2],**

(*2* = reviewers who did not participate in the discussion before interruption.)

---

## **2. Shared Concerns and How They Were Addressed**

### **A. Novelty of Attention–Diversity Hybrid Pruning**

*(zvYn = 2, 2wEG = 2)*

These reviewers asked how our work differs from prior hybrid approaches.

**Clarifications added to the Introduction & Section 4:**

- Prior hybrid methods combine attention/diversity heuristics

    **but do not measure how much diversity they actually retain.**

- Our erank-based study reveals that many “diversity-aware” hybrid methods

    **preserve less diversity than intended. (Sec 4.1)**

- We discovered a new, consistent LVLM pattern **(Sec 4.2)**:
    - **Attention-based pruning** succeeds on *simple* (low-erank) images
    - **Diversity-based pruning** succeeds on *complex* (high-erank) images

        This distinction was not documented in earlier work.

- These empirical insights allow us to improve hybrid methods themselves (BAT [1], VisPruner [2], DivPrune [3] +FasterVLM [4]) **(Sec 4.3)**

**After discussion:**

- No reply before interruption.

---

### **B. Novelty of Empirical Analysis vs. Methodological Contribution**

*(17pH = 6→8, DUt4 = 4→6, 2wEG = 2)*

Reviewers questioned whether the adaptive rule alone suffices as novelty.

We clarified this framing and revised the manuscript accordingly.

**Clarifications added (now highlighted in the revised manuscript):**

**The primary novelty lies in empirical principles, not the mechanism itself:**

Recent work in this area often focuses on uncovering **consistent behavioral phenomena** inside LVLMs and deriving simple, model-agnostic principles from those observations.

- **VTW (AAAI’25 Oral) [5]:** Observes that visual tokens receive almost no attention inside LLM and simply drops all visual tokens after a certain layer.
- **VisionZip (CVPR’25) [6]:** Observes that visual encoders concentrate information into a few highly attended tokens and simply keeps the top-attention tokens while merging the rest.

**Our contribution aligns with this direction by establishing empirical principles that generalize across models.**

1. First **erank-based measurement** of retained diversity across pruning methods **(Sec. 4.1)**
2. Newly identified link between **token diversity and hallucination behaviors (Sec. 4.1)**
3. Image-complexity–dependent preference between attention vs. diversity pruning **(Sec. 4.2)**
4. The adaptive rule is a **minimal proof-of-concept instantiation**, not claimed as a methodological breakthrough
5. Extensive method-agnostic improvements demonstrate that the principle is **robust and fundamental**, not tied to our own method **(Sec. 4.3)**
6. Parameter-free, statistics-driven formulation added **(Sec. 4.3)**

**After discussion:**

- **Resolved:** 17pH (score 6→ 8)
- **Partially resolved:** DUt4 (score 4→6)
- **No reply:** 2wEG

---

### **C. Hyperparameter Sensitivity**

*(17pH = 6→8, 2wEG = 2, g8h8 = 8, DUt4 = 4→6)*

**Added in response:**

- Full sensitivity analysis confirming stable behavior
- **Parameter-free adaptive rule** introduced **(Sec. 4.3)**
- Validation across **LLaVA-7B/13B, LLaVA-NeXT-7B, Qwen2.5-VL-7B** **(Appendix B.1)**

**After discussion:**

- **Resolved:** 17pH, DUt4
- **No reply:** 2wEG, g8h8

---

### **D. Model Diversity**

*(17pH = 6→8, DUt4 = 4→6)*

**Added new evaluations:**

- **LLaVA-1.5-13B**
- **LLaVA-NeXT-7B**
- **Qwen2.5-VL-7B**

All models consistently showed the same empirical pattern and benefited from the adaptive strategy **(Appendix B.1).**

**After discussion:**

- **Resolved.**

---

### **E. Efficiency Analysis**

*(17pH = 6→8, DUt4 = 4→6)*

**Added details in Appendix A:**

- end-to-end latency
- memory usage
- erank overhead (≈3.2% of runtime)
- comparison with in-LLM pruning methods

**After discussion:**

- **Resolved.**

---

---

### Meta-Review · Area_Chair_KsB9 · 2026-01-07

**Summary:**

This paper presents a systematic empirical study of visual token pruning in large vision–language models, focusing on the interaction between attention-based and diversity-based strategies. Using effective rank (erank) and attention entropy as diagnostic tools, the authors uncover consistent patterns linking image complexity, retained token diversity, and hallucination behavior. Based on these insights, they derive a simple, model-agnostic adaptive pruning rule that improves multiple existing methods across a wide range of benchmarks and backbones.

Reviewers raised concerns primarily around `the novelty of the contribution`, questioning whether the work goes beyond combining existing heuristics, as well as requests for `broader model validation`, `robustness analysis`, `efficiency evaluation`, and `clarification of design choices`. During rebuttal and discussion, the authors provided extensive additional experiments, clarified the intended empirical contribution, and demonstrated that the discovered principles consistently improve diverse pruning methods. Taken together, the concerns were sufficiently addressed, and the paper makes a strong empirical contribution that is valuable to the community.

**Reviewer Concerns:**

**1. [Addressed] Limited algorithmic novelty relative to prior hybrid pruning methods
(by: `zvYn`, `2wEG`, partially `17pH`)**

Several reviewers questioned whether the proposed adaptive pruning mechanism constitutes sufficient novelty, noting that prior work (e.g., BAT, VisionZip, PruMerge+) already combines attention and diversity signals, and that the adaptive threshold itself appears heuristic rather than algorithmically novel.

The rebuttal clarified that the core contribution is empirical rather than architectural, emphasizing: (i) the first erank-based quantification of retained token diversity across pruning methods, (ii) the newly identified relationship between diversity and hallucination behavior, and (iii) the image-complexity–dependent preference between attention- and diversity-based pruning. Additional experiments demonstrated that these principles consistently improve existing pruning methods, addressing concerns that the work is merely a re-combination of prior ideas.

**2. [Addressed] Generalization beyond a single LVLM backbone
(by: `17pH`, `DUt4`)**

Reviewers expressed concern that most results were initially based on LLaVA-1.5-7B, raising questions about whether the empirical findings generalize to other architectures.

The authors added extensive evaluations on LLaVA-1.5-13B, LLaVA-NeXT-7B, and Qwen2.5-VL-7B, showing consistent trends and improvements across architectures. Reviewers acknowledged that these additions sufficiently addressed the concern.

**3. [Addressed] Insufficient robustness, sensitivity, and failure-mode analysis
(by: `17pH`, `DUt4`, `2wEG`)**

Concerns were raised regarding the stability of the erank-based adaptive rule under noisy or corrupted inputs, the sensitivity to hyperparameters, and the lack of systematic failure analysis.

The rebuttal added COCO-C robustness experiments, hyperparameter sensitivity studies, a parameter-free adaptive formulation, and expanded qualitative failure analyses. These additions directly addressed the reviewers’ requests.

**4. [Addressed] Incomplete system-level efficiency evaluation
(by: `17pH`, `DUt4`)**

Some reviewers noted that the original submission focused mainly on FLOPs reduction, lacking concrete measurements of latency and memory usage.

The authors added detailed end-to-end latency and GPU memory evaluations, including comparisons with in-LLM pruning methods (e.g., PDrop, SparseVLM), demonstrating practical efficiency gains. This concern was resolved.

**5. [Partially Addressed] Scope and positioning of the empirical contribution
(by: `2wEG`)**

One reviewer remained skeptical that the empirical insights alone justify acceptance, viewing the adaptive rule as incremental despite added analysis.

While the rebuttal clarified the paper's intended positioning and strengthened the empirical evidence, this concern reflects a difference in valuation rather than a missing technical response.

**Reviewer Scores:**

**Reviewer 17pH (6 -> 8)**

Additional experiments and clarifications on robustness, efficiency, and generalization resolved the reviewer’s concerns, supporting a clear upward revision.

**Reviewer DUt4 (4 -> 6)**

Multi-backbone validation and strengthened efficiency analyses addressed the reviewer’s main concerns, justifying a moderate increase.

**Reviewer g8h8 (8 -> 8)**

Positive throughout; the rebuttal reinforced an already favorable assessment.

**Reviewer zvYn (2 -> 4)**

Clarification of the empirical contribution and broader validation mitigated novelty concerns to some extent, warranting a partial upward revision.

**Reviewer 2wEG (2 -> 3)**

While still skeptical about algorithmic novelty, the strengthened empirical evidence supports a modest increase.

---

### Decision · Program_Chairs · 2026-01-26

Accept (Poster)